# A CRISPR-Cas9 screen identifies EXO1 as a formaldehyde resistance gene

Yuandi Gao [1,3], Laure Guitton-Sert [1,3], Julien Dessapt[1], Yan Coulombe[1], Amélie Rodrigue[1], Larissa Milano [1], Andréanne Blondeau[1], Nicolai Balle Larsen [2], Julien P. Duxin[2], Samer Hussein[1], Amélie Fradet-Turcotte [1] & Jean-Yves Masson [1] ✉

Fanconi Anemia (FA) is a rare, genome instability-associated disease characterized by a deficiency in repairing DNA crosslinks, which are known to perturb several cellular processes, including DNA transcription, replication, and repair. Formaldehyde, a by-product of metabolism, is thought to drive FA by generating DNA interstrand crosslinks (ICLs) and DNA-protein crosslinks (DPCs). However, the impact of formaldehyde on global cellular pathways has not been investigated thoroughly. Herein, using a pangenomic CRISPR-Cas9 screen, we identify EXO1 as a critical regulator of formaldehyde-induced DNA lesions. We show that EXO1 knockout cell lines exhibit formaldehyde sensitivity leading to the accumulation of replicative stress, DNA double-strand breaks, and quadriradial chromosomes, a typical feature of FA. After formaldehyde exposure, EXO1 is recruited to chromatin, protects DNA replication forks from degradation, and functions in parallel with the FA pathway to promote cell survival. In vitro, EXO1-mediated exonuclease activity is proficient in removing DPCs. Collectively, we show that EXO1 limits replication stress and DNA damage to counteract formaldehyde-induced genome instability.

The cellular genome is constantly exposed to various forms of DNA damage. DNA-protein crosslinks (DPCs) are common lesions characterized by the covalent association of proteins to DNA, which can stall DNA transactions that must slide through DNA molecules. Enzymatic DPCs are produced by faulty enzymatic reactions, such as the abortion of topoisomerase 1/2 cleavage complexes in the presence of camptothecin or etoposide[1,2]. Non-enzymatic DPCs are produced after exposure to UV[3] or aldehydes[4]. Aldehydes, present in the environment such as in urban areas, are also produced during physiological cell and alcohol metabolism. Formaldehyde is a typical aldehyde naturally generated by every living organism[5]. As a metabolic intermediate, it is required for the biosynthesis of purines and certain amino acids.

Determination of endogenous levels of formaldehyde in human blood by several methods[6–8] revealed high concentrations with plasma levels reaching $100\,\mu M$[6]. Primary sources of formaldehyde include histone demethylation and dealkylation of methylated DNA, which leads to the accumulation of DPCs. DPC repair processes involve tyrosyl-DNA phosphodiesterase 1 and 2 (TDP1/2)[9,10], which cleave 3′ and 5′ tyrosyl-DNA crosslinks, SPARTAN protease-mediated or proteasome-mediated DPC cleavage at stalled replication forks[11–13], and MRN-CtIP to initiate endonucleolytic cleavage of DPCs at double-strand breaks (DSBs)[14].

Formaldehyde also generates DNA interstrand crosslinks (ICLs) via a methylene bridge formed between the exocyclic amino groups of

[1]CHU de Québec Research Center, Oncology Division, Department of Molecular Biology, Medical Biochemistry and Pathology, Laval University Cancer Research Center, 9 McMahon, Québec City, QC G1R 3S3, Canada. [2]Novo Nordisk Foundation Center for Protein Research, Faculty of Health and Medical Sciences, University of Copenhagen, 2200 Copenhagen, Denmark. [3]These authors contributed equally: Yuandi Gao, Laure Guitton-Sert. ✉e-mail: Jean-Yves.Masson@crchudequebec.ulaval.ca

adjacent DNA bases[15]. The removal of ICLs usually is managed by the Fanconi Anemia (FA) pathway. FA is a rare recessive genetic disorder characterized by congenital abnormalities, bone marrow failure and cancer predisposition[16]. It is caused by mutations in any of the 22 FA genes identified so far (FANCA-FANCW), which lead to a deficiency in repairing ICLs. The FA pathway uses the ubiquitin E3 ligase FANCL to catalyze the monoubiquitination of the FANCI-FANCD2 (ID2) hetero-dimer complex following the accumulation of ICLs or replicative stress. The monoubiquitinated ID2 complex associates with damaged DNA and orchestrates ICL removal and subsequent DNA repair by nucleases/repair proteins SLX4/FANCP, BRCA1/FANCS, BRCA2/FANCD1, BACH1/FANCJ, and PALB2/FANCN[16]. Incisions on either side of the ICL lead to the formation of a DSB, which is then repaired by homologous recombination (HR). HR begins with a resection step by BRCA1-CtIP[17] and the MRN-RPA-BLM-EXO1/DNA2 complex[18], leading to single-strand DNA that is rapidly coated by RPA. Then, the BRCA1/BRCA2/PALB2 complex allows the replacement of RPA by RAD51, which forms a nucleoprotein filament required for the invasion of the sister chromatid and subsequent DNA repair[19].

Recently, aldehydes have been described to participate in FA development and severity in FA mouse models. A direct connection between aldehydes and FA was first established as FANCD2-, FANCD1-deficient DT40 cells, as well as FANCC-, FANCG-deficient human RKO cells, were found to be hypersensitive to plasma-level formaldehyde[20]. Furthermore, the inactivation of Aldehyde dehydrogenase 2 (ALDH2), which oxidizes acetaldehyde to acetic acid, exacerbates FA phenotypes[21,22]. Studies revealed that *Aldh2*−/− *Fancd2*−/− double-mutant mice carried subtle defects such as kinked tails and eye defects and perished within 3−6 months because of an acute illness akin to acute lymphoblastic leukemia[22]. It was also reported that aged *Aldh2*−/− *Fancd2*−/− mice that do not develop leukemia suffer from aplastic anemia with an accumulation of DNA damage in the hematopoietic stem and progenitor cell pool[23]. Alcohol dehydrogenase 5 (ADH5) later emerged as another suppressor of formaldehyde toxicity. Studies in induced pluripotent stem cells (iPSCs) highlighted that ADH5 is the primary defense against formaldehyde while ALDH2 acts as a backup enzyme in such background[24]. These studies suggested that FA is exacerbated by aldehyde-mediated genotoxicity and protected by the ADH5/ALDH2 enzymes.

Genes that respond to formaldehyde have not been investigated thoroughly. Therefore, to chart new pathways underlying the biological response to formaldehyde, we performed a pangenomic CRISPR-Cas9 screen. Herein, we provide evidence that Exonuclease 1 (EXO1) orchestrates the removal and repair of both formaldehyde-induced DPCs and ICLs to prevent genome instability. Functional characterization of the genes that respond to formaldehyde may help to better understand the exact mechanism by which this damaging compound participates in the etiology of FA.

## Results
### Identification of EXO1 as a formaldehyde response gene
To identify genes involved in response to formaldehyde, we performed a pangenomic CRISPR-Cas9 screen in h-TeRT-immortalized RPE-1 cells (RPE-1), a well-characterized normal cell line commonly used for CRISPR screen (Fig. 1a, see "Methods" for details). RPE-1 cells stably expressing Cas9 were infected with TKO_v1 CRISPR library[25]. Infected cells were then divided into two conditions, either untreated or treated with formaldehyde at 70 μM, a dose that kills ~80% of the cells in 15 days (Fig. 1a; Supplementary Fig. 1a). Cells were collected before and after 9 and 15 days of treatment in both untreated and treated conditions. Variation in sgRNA between untreated and treated conditions at each time point was analyzed using DrugZ, a program developed for CRISPR screen analysis[26]. Normalized gene-level *Z*-scores (normZ) represented in Supplementary Fig. 1b are available in Supplementary Data 1. We then focused our attention on drop out

results. Genes for which normZ was lower than −3 after 15 days of treatment were first considered as significantly decreased. Several data validated the reliability of our screen. Gene ontology of biological processes using DAVID online tool showed that processes involved in formaldehyde catabolism and DNA damage response were enriched (Fig. 1b, c). We then narrow down our gene list by selecting sgRNA that were consistently and significantly decreased over time (normZ lower than −3 on both Day 9 and Day 15, FDR < 0.05). A list of 20 genes was selected (Fig. 2a). ADH5 and ESD, two enzymes involved in formaldehyde catabolic processes[27], were classified as top hits (green dots). Furthermore, FA pathway genes and transcription-coupled nucleic excision repair (TC-NER, blue and yellow dots, respectively) such as CSB/ERCC6 were enriched, as previously described[27,28] (Figs. 1b, 2a and Supplementary Fig. 1c). This corroborates new findings showing that endogenous formaldehyde impedes transcription and that CSB/ERCC6 protects the kidney and brain against endogenous formaldehyde[29].

Then, we validated these individual 20 hits from our screen using siRNA-mediated depletion in the presence of formaldehyde at a concentration that gave the highest difference between siCTL conditions and two siRNA targeting ADH5 and FANCA (Fig. 2b, Supplementary Fig. 2a). Knockdown of the EXO1 gene led to ~40% cell death compared to the control condition (Fig. 2b, Supplementary Fig. 2b). Interestingly, sensitivity to formaldehyde after EXO1 depletion was already observed in two previous CRISPR-Cas9 screens, strengthening our results[27,28] (Supplementary Fig. 1c). EXO1 is a versatile 5′ → 3′ exonuclease and a DNA structure-specific DNA endonuclease involved in DNA mismatch repair (MMR), DNA DSB repair, and DNA replication stress[30] (Fig. 2c). EXO1 inactivation by CRISPR-Cas9 strategy in two clones (KO7 and 11, Fig. 2d) also led to formaldehyde sensitivity. This was rescued by complementation of EXO1 KO11 with a single copy of EXO1 introduced in the safe harbor site AAVS1[31] (Fig. 2e, f), confirming that formaldehyde sensitivity was due to EXO1 depletion. As ADH5 is the main enzyme in charge of formaldehyde catabolism, and to a lesser extent ALDH2, we confirmed that the levels of expression of ADH5/ALDH2 were not affected by the absence of EXO1 (Supplementary Fig. 2c). While EXO1 and DNA2 share redundancy in the HR step in replication-coupled repair[32], the depletion of DNA2 did not lead to sensitivity to either formaldehyde or mitomycin C (MMC) (Supplementary Fig. 2d). These results suggest that the function of DNA2 in formaldehyde and MMC sensitivity is negligible.

### EXO1 deletion exacerbates formaldehyde-induced DNA damage in S-phase
The FA pathway is required for the DNA replication stress response, for instance, after MMC treatment or a low dose of aphidicolin[33]. To investigate whether formaldehyde was also inducing DNA replicative stress, we first analyzed the phosphorylation of Chk1 on serine 345, which is catalyzed by ATR when DNA replication is perturbed[34]. Formaldehyde-induced pChk1(S345) was heightened significantly in the two EXO1 KO clones (Fig. 3a (upper panel)). The increase in pChk1(S345) appeared dependent on EXO1 as it was reversed in complemented KO cell line (Fig. 3a (lower panel)). These data suggested that EXO1 limits DNA replicative stress induced by formaldehyde.

We also analyzed whether replication dynamics were perturbed under formaldehyde treatment by DNA combing strategy after 30 min of incorporation of IdU, followed by CldU, at the end of formaldehyde treatment (18 h). First, we checked that depletion of EXO1 does not strongly perturb fork progression in untreated conditions by measuring IdU track lengths in the absence of formaldehyde treatment (Supplementary Fig. 3b). Then, consistent with a DNA replication stress response, formaldehyde caused a decrease in CldU track length even in wild-type conditions (Fig. 3b, Supplementary Fig. 3a). Moreover, under formaldehyde treatment, EXO1 depletion induced a significant

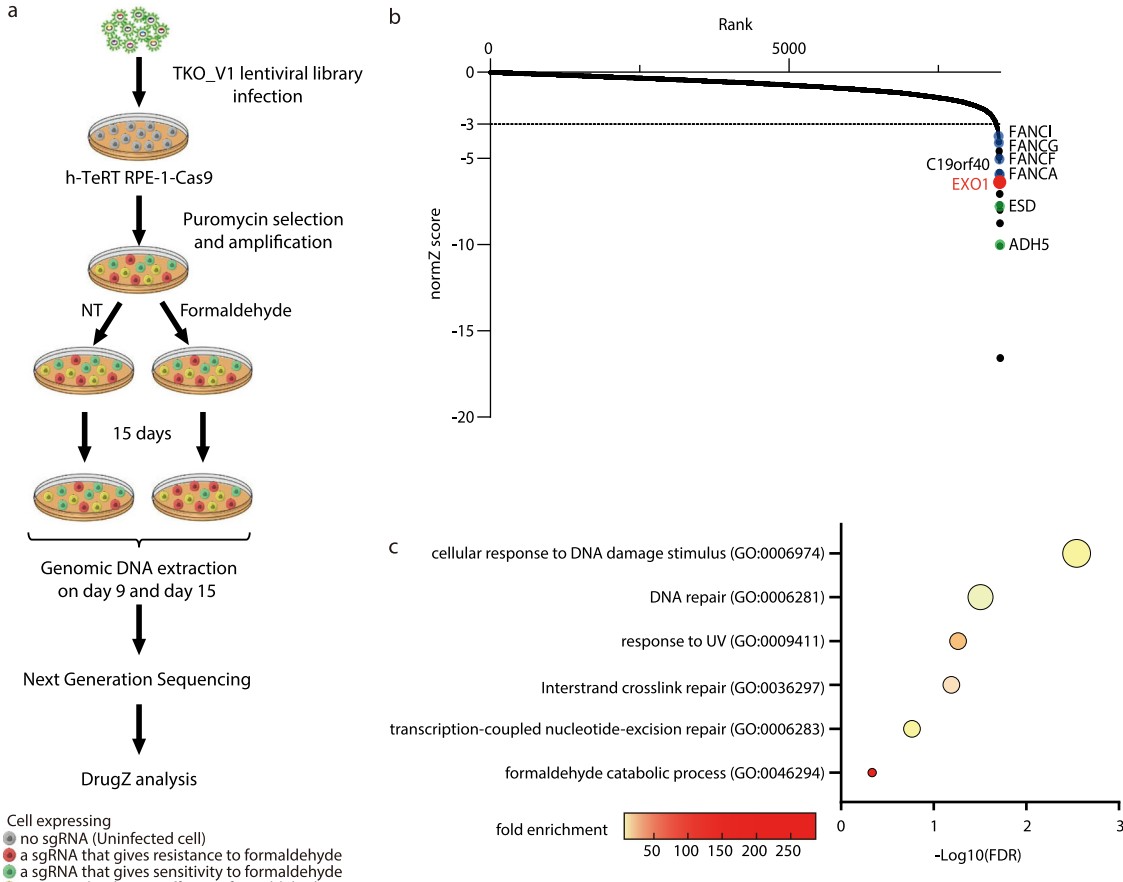

**Fig. 1 | A CRISPR-Cas9 screen to identify genes that are responding to formaldehyde. a** Schematic overview of the formaldehyde CRISPR-Cas9 screen. In brief, RPE-1 cells were infected by the TKO_V1 lentiviral library, followed by puromycin selection and amplification. Infected cells were continuously treated with or without 70 μM formaldehyde and genomic DNA extraction was carried out on day 9 and day 15 post-treatment. Next-generation sequencing results were analyzed with DrugZ[26]. NT: untreated cells. **b** Normalized gene-level *Z*-scores (normZ score) for genes that appeared in results from the negative selection of the CRISPR-Cas9 screen. The dashed line corresponds to the significance threshold (−3 represents a probability value of less than 0.001). Genes in blue belong to the FA pathway, in green to formaldehyde catabolic process and EXO1 is in red. **c** Gene ontology of biological processes corresponding to genes targeted by sgRNA that are significantly decreased after formaldehyde treatment presented by DAVID software. The size of the circle is proportional to the number of genes identified by the screen included in the indicated biological processes. FDR false discovery rate.

decrease in CldU track length that was rescued by complementation with EXO1 WT. To better detail replication fork dynamics as a function of EXO1 status, we performed DNA fiber analyses to assess nascent strand degradation. As a control, we used BRCA2 knockdown, which led to nascent strand degradation in the presence of HU, as reported previously (Supplementary Fig. 3c). We incorporated both IdU followed by CldU and then challenged cells with formaldehyde for 4 h to monitor if CldU tracks length decreased. Again, in untreated conditions, there was no difference in IdU track length distribution between WT and EXO1 depleted cells, showing that EXO1 depletion does not impact replication track length by itself (Supplementary Fig. 3d–e). Conversely, EXO1 inactivation in untreated cells led to a slight increase in strand degradation, which was enhanced further by formaldehyde treatment (Fig. 3c, Supplementary Fig. 3d) indicating a protective role for EXO1 in DNA replication fork degradation. In addition, these results indicate that EXO1 plays a role at DNA replication forks to prevent DNA replication stress and replication fork degradation in the presence of formaldehyde.

Impediments to DNA replication can lead to fork collapse and the formation of DSBs[35]. We therefore analyzed whether formaldehyde induced DNA damage by monitoring γH2AX foci formation. A dose-dependent increase of γH2AX foci per nuclei occurred with increasing doses of formaldehyde (Fig. 4a). Interestingly, DSBs induction was mainly observed in S-phase, as EdU positive cells displayed a higher level of γH2AX foci formation than EdU negative cells (Fig. 4b, Supplementary Fig. 4a), supporting the hypothesis that formaldehyde-induced DNA damage is linked to DNA replication. As DSBs are mainly repaired by HR in S-phase, we assessed immunofluorescence foci of two proteins recruited to the chromatin during HR, RPA and RAD51, after formaldehyde treatment. For RAD51 and RPA, the number of foci per nuclei increased significantly after 100 μM of formaldehyde (Fig. 4c, d), suggesting that the HR DNA repair pathway is engaged. Nevertheless, staining of phospho-53BP1 (S1778) and phospho-DNA-PKcs (S2056) shows that DNA repair through non-homologous end joining (NHEJ) is also activated, but to a lesser extent (Supplementary Fig. 4b).

Having established that formaldehyde exposure induces DNA damage mostly in S-phase, leading to predominant DSB repair by HR, we evaluated the impact of EXO1 depletion on γH2AX foci formation in this phase. Remarkably, after formaldehyde treatment, γH2AX staining in EdU positive cells was increased in the absence of EXO1 (Fig. 5a) while being rescued by complementation with EXO1 WT (Supplementary Fig. 4c). Conversely, EXO1 depletion did not significantly affect γH2AX foci formation after ionizing radiation (IR) (Fig. 5a). The latter observation is consistent with the absence of significant IR sensitivity in mouse EXO1 KO cells[36] and human EXO1 KO clones compared to WT cells (Supplementary Fig. 5a). This might be explained by the fact that DNA2 can compensate for the absence of EXO1[32]. The elevated

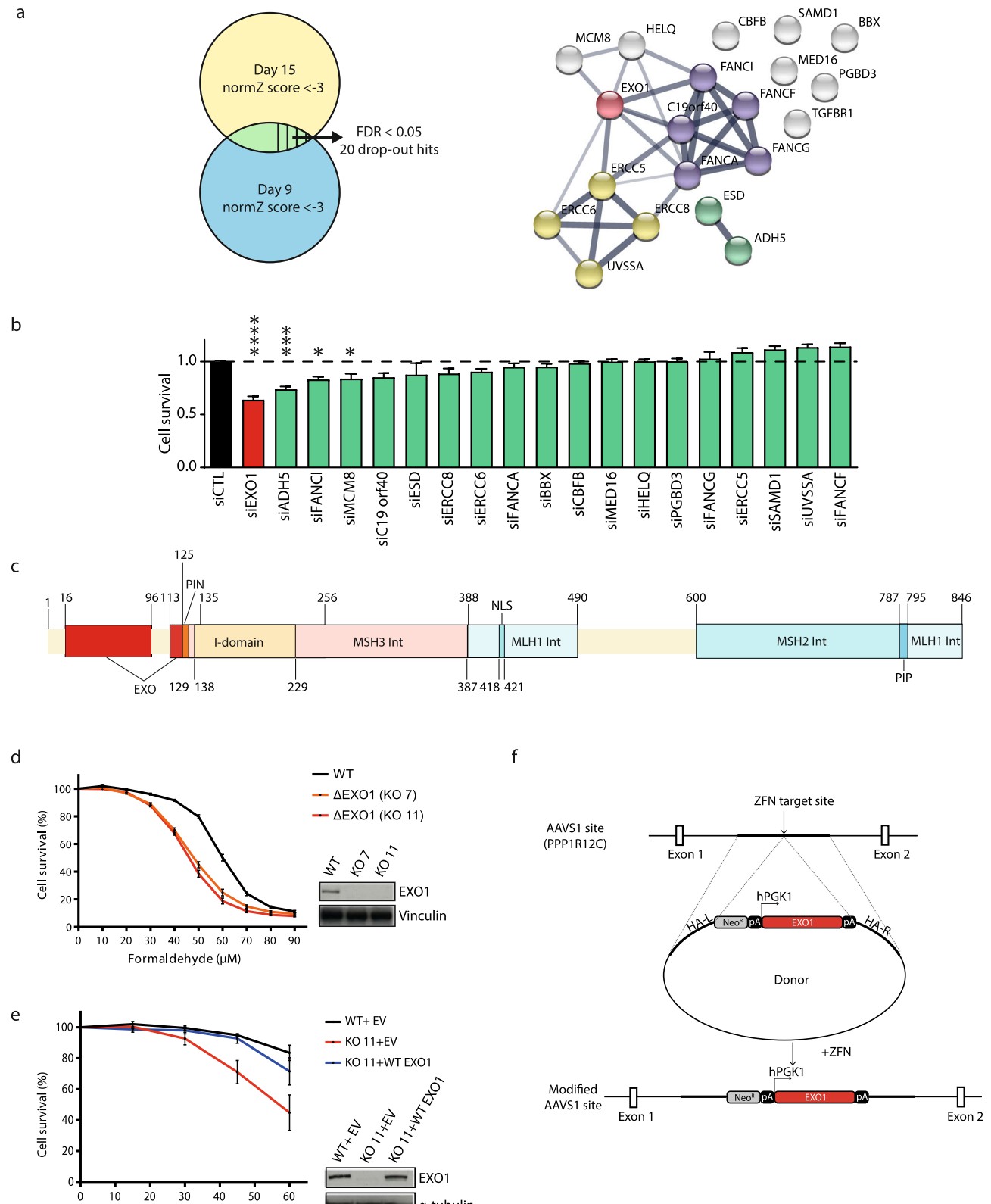

γH2AX staining in EXO1-depleted cells exposed to formaldehyde suggested an accumulation of DSBs. We assessed if these DSBs were dependent on the structure-specific nuclease Mus81, which promotes the conversion of ICLs to DSBs in S-phase[37]. Indeed, the knockdown of Mus81 in EXO1-depleted cells led to a decrease in the number of γH2AX foci (Fig. 5c, d). Thus, knockdown of Mus81 is required to avoid the accumulation of DSBs and fork collapse after replication perturbation by formaldehyde when EXO1 is depleted. We then assessed whether

HR/NHEJ markers were affected by EXO1 deletion under formaldehyde treatment. Indeed, RPA, RAD51, and p53BP1 (S1778) foci were all increased (Fig. 6a). This contrasted with IR, where EXO1 deletion had a limited impact on RAD51 and p53BP1 (S1778) foci (Supplementary Fig. 5b). Hydroxyurea-treated EXO1 deficient cells did not elicit such effects on DNA damage (Supplementary Fig. 6a). Consistent with an increase in the number of RPA foci, exposure of EXO1 knockout cells to formaldehyde led to enhanced RPA S4/8 phosphorylation, a marker

**Fig. 2 | A secondary knockdown screen identifies EXO1 as a formaldehyde resistance gene. a** Left. 20 drop-out hits were selected by their normZ score (lower than −3 on both day 9 and day 15, and a FDR lower than 0.05). Right. The 20 drop-out hits are presented by STRING. Green: formaldehyde catabolic processes, blue: FA pathway genes, yellow: transcription-coupled nucleotide excision repair genes, red: EXO1, gray: other genes non-assigned to any specific pathway. **b** Cell survival in RPE-1 cells transfected with siCTL or individual siRNAs targeting the 20 drop-out hits from the CRISPR-Cas9 screen. Data are presented with ±SEM from three independent experiments. *$p < 0.1$, ***$p < 0.001$ and ****$p < 0.0001$ (ordinary one-way ANOVA). **c** Schematic overview of domains in the human EXO1 protein. **d** Survival curve of RPE-1 EXO1 wild-type (WT) cells and RPE-1 EXO1 knockout clones

(KO 7, KO 11) treated with different concentrations of formaldehyde for 96 h. Data are presented with ±SEM from three independent experiments. Protein level of EXO1 in RPE-1 WT, KO 7, KO 11 cells, with vinculin as loading control. **e** Survival curve of RPE-1 EXO1 WT with AAVS1 empty vector (WT + EV) cells, KO 11 with AAVS1 empty vector (KO 11 + EV) cells and KO 11 complemented with WT EXO1 (KO 11 + WT EXO1) cells treated with different concentrations of formaldehyde for 96 h. Data are presented with ±SEM from three independent experiments. Protein level of EXO1 in RPE-1 AAVS1 complemented cells, with α-tubulin as loading control. **f** Scheme for generating single copy of EXO1 WT complemented cells using integration at the AAVS1 single locus.

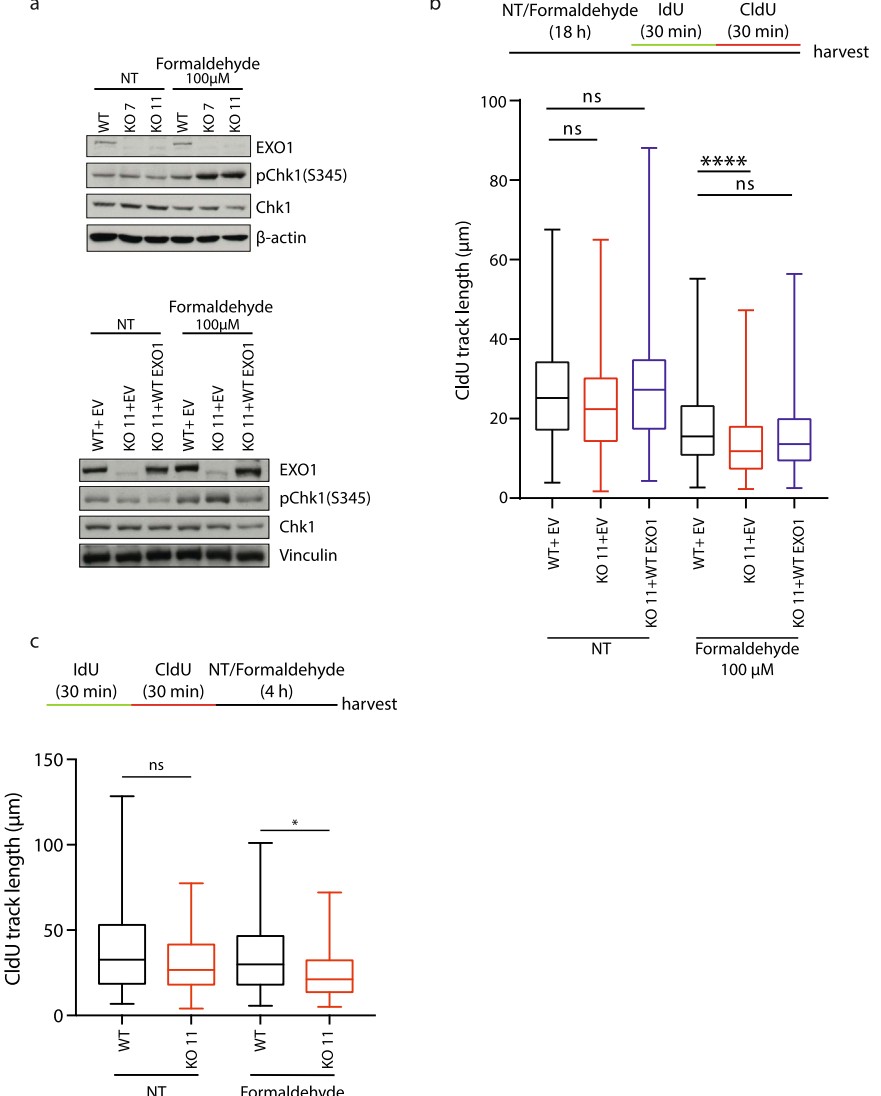

**Fig. 3 | EXO1 limits DNA replication stress and DNA damage induced by formaldehyde. a** Protein levels of EXO1, pChk1 (pSer 345), and Chk1 in RPE-1 WT, KO 7, KO 11, and AAVS1 complemented cells with or without 100 μM formaldehyde treatment for 18 h, with housekeeping genes (β-actin or vinculin) as loading controls. **b** CldU track length of DNA fibers from AAVS1 complemented cells with or without 100 μM formaldehyde treatment for 18 h. Data are shown with mean ± SEM from three independent experiments. ns: non-significant, ****$p < 0.0001$ (one-way ANOVA, followed by Kruskal–Wallis test). EV: AAVS1 + Empty Vector. **c** CldU track length of DNA fibers in RPE-1 WT and EXO1 KO 11 cells with the indicated treatment. Data are shown with mean ± SEM from three independent experiments. ns: non-significant, *$p < 0.1$. (One-way ANOVA, followed by Kruskal–Wallis test).

for the induction of the DNA damage response and DNA resection (Fig. 6b).

These data hinted at the recruitment of EXO1 to formaldehyde-induced DNA damage. First, subcellular fractionation experiments showed that EXO1 binds chromatin after an

acute formaldehyde treatment (Fig. 6c, Supplementary Fig. 6b). Moreover, when recruited to DNA after replicative stress or DSBs, EXO1 is phosphorylated on S714 by ATM in response to DSBs and by ATR following replication stress[38–40]. Interestingly, we observed a substantial increase in the number of pEXO1

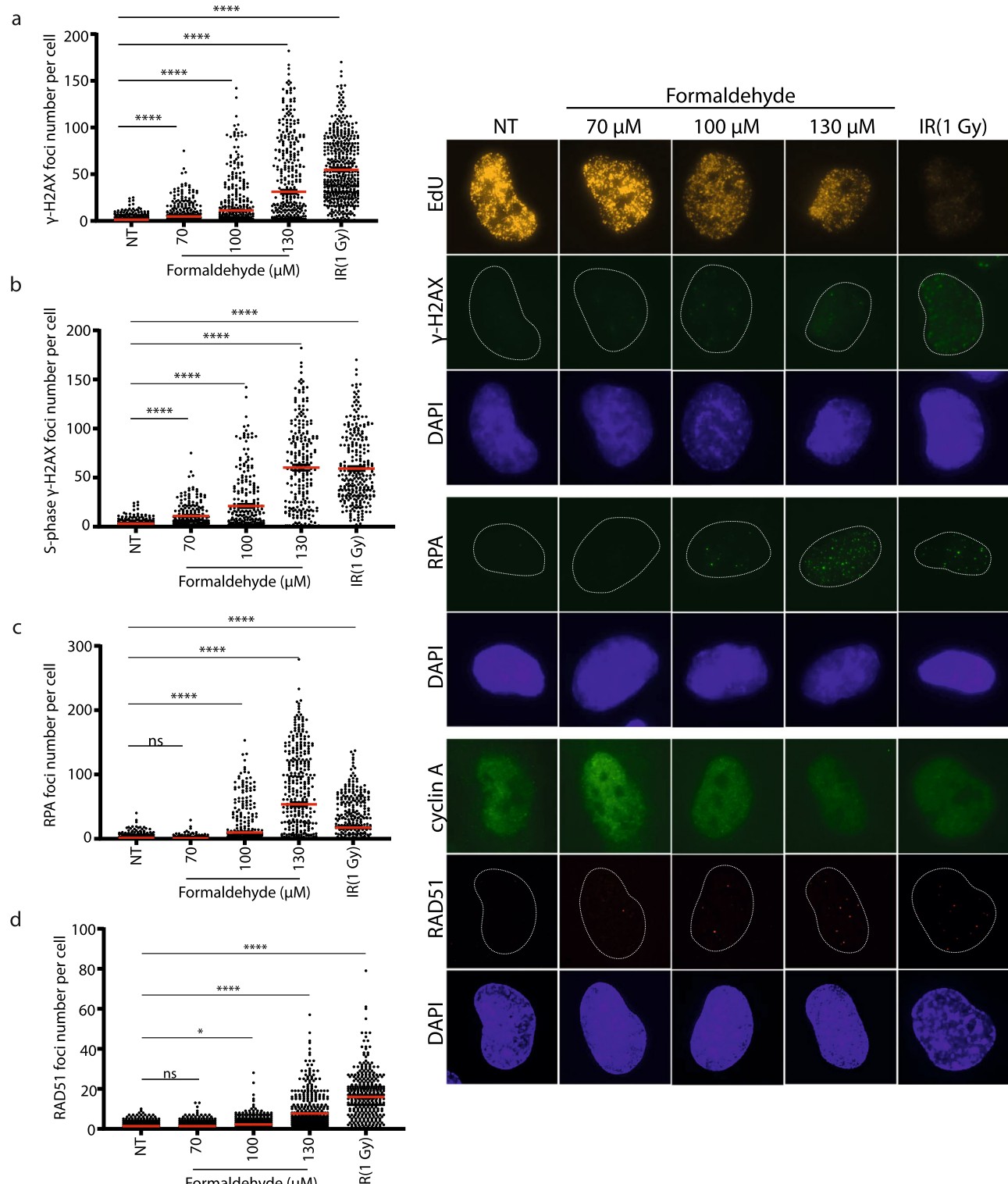

**Fig. 4 | Formaldehyde induces DNA double-strand breaks (DSBs) mainly in S/G2 phase and homologous recombination.** Immunofluorescence (IF) staining against **a** γH2AX (pSer 139), **b** γH2AX (pSer 139) in S-phase cells determined by EdU staining, **c** RPA2, and **d** RAD51 as well as cyclin A in RPE-1 cells under 0, 70, 100, 130 μM formaldehyde treatment for 18 h or 1 Gy irradiation. Data are shown with mean ± SEM from three independent experiments. RAD51 foci were quantified in cyclin A positive cells. ns: non-significant, *$p < 0.1$, ****$p < 0.0001$ (one-way ANOVA, followed by Kruskal–Wallis test). Representative images of IF are shown on the right.

(S714) foci in S-phase after formaldehyde treatment (Fig. 6d), the majority of which colocalized with γH2AX (Supplementary Fig. 6c). These data indicate that EXO1 is particularly important in combating DNA lesions generated by formaldehyde in S-phase.

## EXO1 acts on ICLs and DPCs

The observed accumulation of DSBs and replication stress under formaldehyde treatment could originate from two major types of DNA damage, ICLs and DPCs. We first scrutinized if EXO1 could act at ICLs as its deletion led to sensitivity to MMC, which mainly

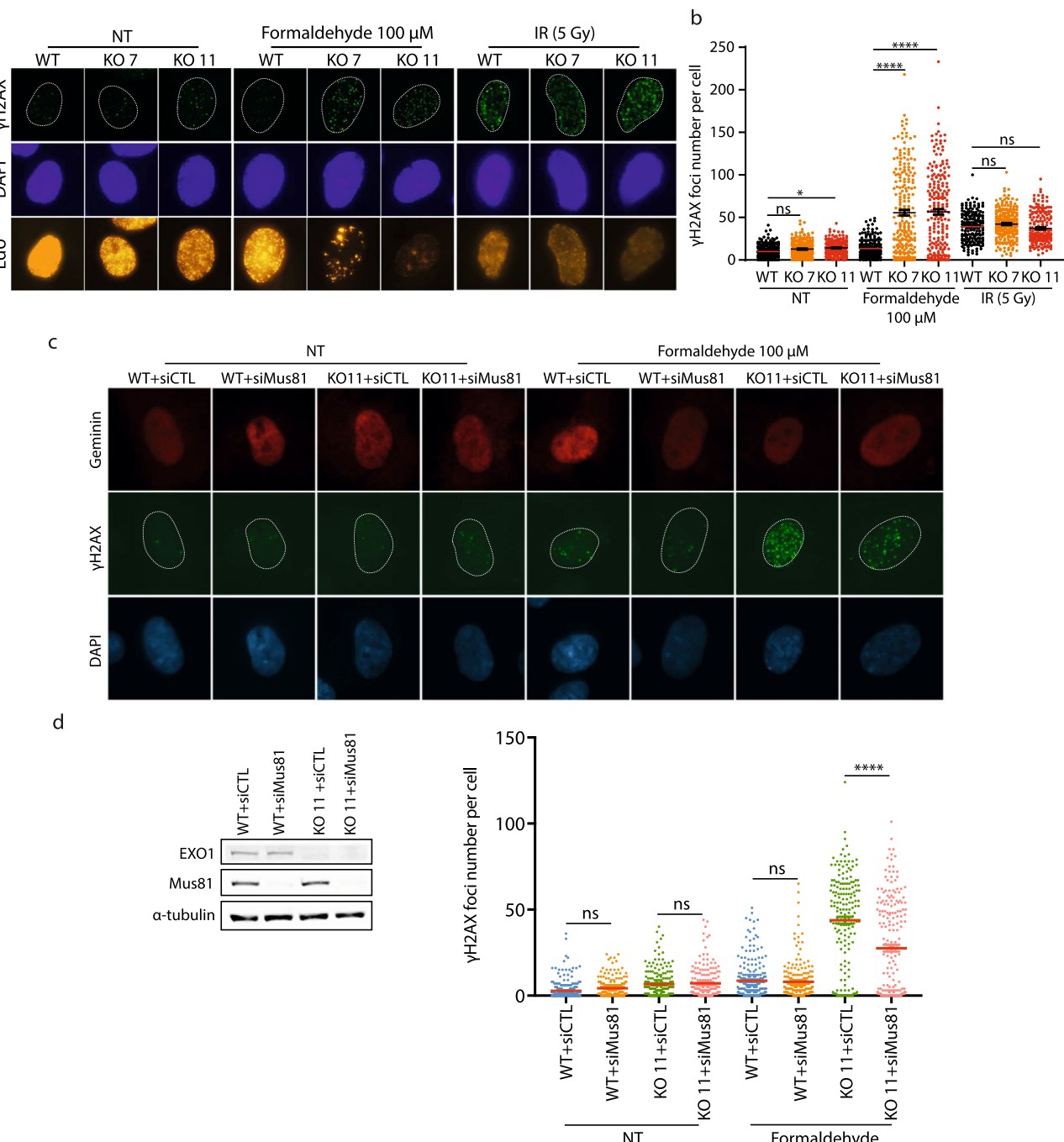

**Fig. 5 | EXO1 responds to formaldehyde-induced DSBs. a** IF staining against γH2AX (pSer 139) in S-phase indicated by EdU staining in RPE-1 WT and EXO1 KO (KO 7, KO 11) cells without treatment or treated with 100 μM formaldehyde for 18 h or 5 Gy irradiation. **b** Quantification of γH2AX (pSer 139) foci in S-phase cells for **a**. Data are shown with mean ± SEM from three independent experiments. ns: non-significant, *$p < 0.1$, ****$p < 0.0001$ (one-way ANOVA, followed by Kruskal–Wallis test). **c** IF staining against γH2AX (pSer 139) in S/G2-phase indicated by Geminin staining in RPE-1 WT and EXO1 KO 11 cells transfected with siCTL or siMus81, treated with or without 100 μM formaldehyde for 18 h. **d** Left. Protein levels of EXO1, Mus81 in RPE-1 WT and EXO1 KO 11 cells transfected with siCTL or siMus81, with α-tubulin as a loading control. Right. Quantification of γH2AX (pSer 139) foci in S/G2-phase cells for **c**. Data are shown with mean ± SEM from three independent experiments. ns: non-significant, ****$p < 0.0001$ (one-way ANOVA, followed by Kruskal–Wallis test).

induces ICL lesions (Fig. 7a). To visualize the potential recruitment of EXO1 to ICLs, we took advantage of a live cell imaging assay specifically developed to study protein recruitment to ICLs in cellulo[41,42] and based on the induction of localized photo-activated DNA crosslink through UV laser irradiation after trimethylpsoralen (TMP) treatment. As EXO1 is recruited to UV laser-induced DNA damage, we first set the irradiation conditions that allowed the recruitment of UHRF1, a protein shown to be recruited at UV irradiated sites specifically in TMP pre-treated cells[42] (Supplementary Fig. 7a), but without inducing the recruitment of transiently expressed GFP-EXO1 in the absence of TMP. Using these conditions, we found that GFP-EXO1 is rapidly recruited to the laser-activated site after TMP pre-treatment, suggesting a role in ICLs cellular response (Fig. 7b). The exonuclease-dead mutant GFP-EXO1 D173A was recruited with the same kinetics as the WT, suggesting that EXO1 nuclease activity is

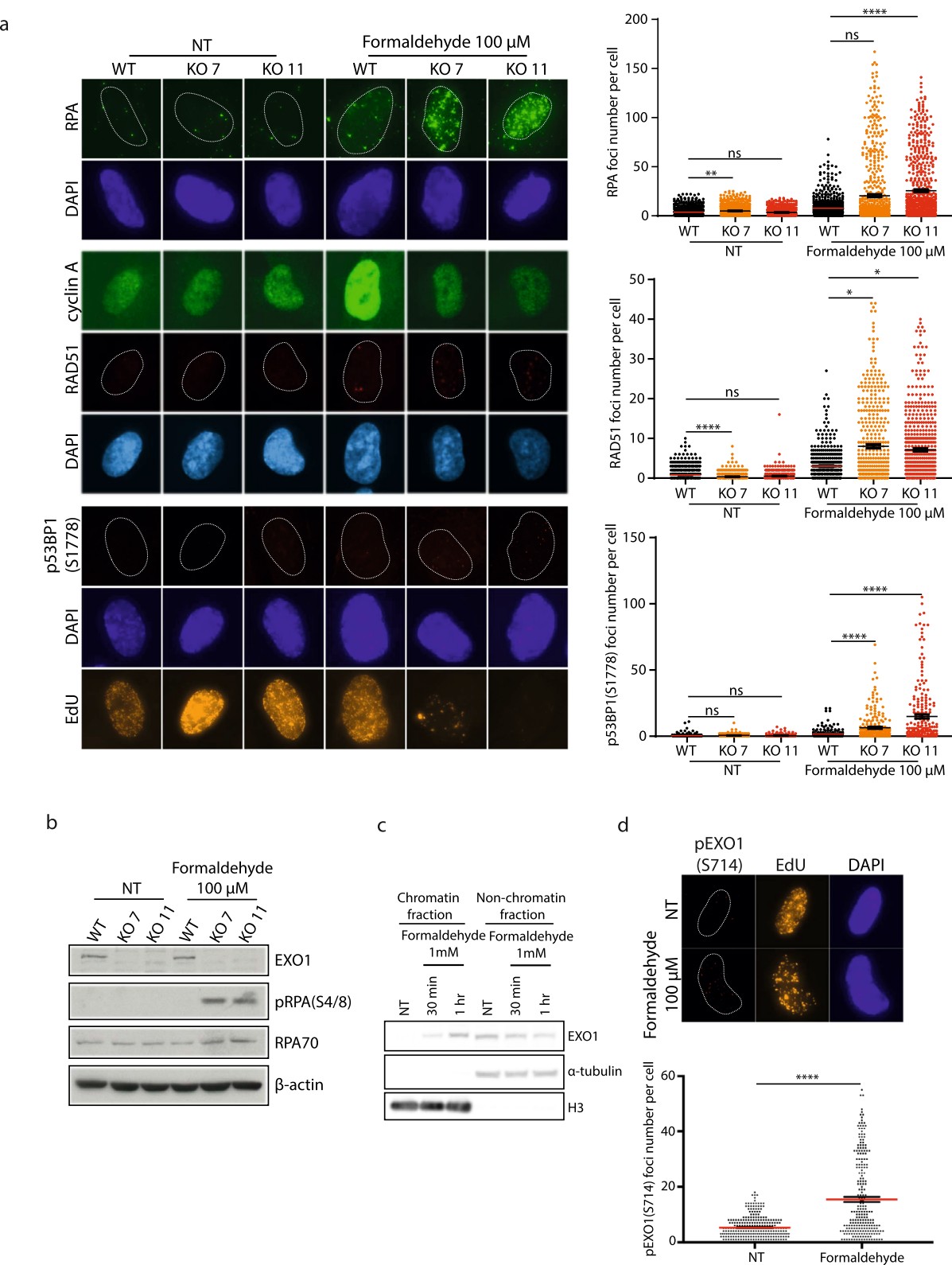

dispensable for its recruitment (Fig. 7b). We also monitored the recruitment of EXO1 following treatment with angelicin, a photoreactive psoralen analogue, which unlike TMP can form only monoadducts and not ICLs upon UV irradiation. The irradiation conditions were set using CSB as control, a protein known to accumulate at sites of angelicin monoadducts[43]. Under conditions where CSB was recruited in the presence of angelicin and not with the laser alone, none of the cells showed EXO1 recruitment,

suggesting EXO1 responds to ICLs rather than monoadducts (Supplementary Fig. 7b).

The second type of damage induced by formaldehyde is DPCs. We hypothesized that EXO1 could prevent DNA damage after formaldehyde treatment by clearing DPCs through degradation of DNA crosslinked to proteins. First, we monitored the accumulation of DPCs in EXO1 KO cells challenged with formaldehyde by immunodetection of human topoisomerase I-DNA covalent complexes (TOP1cc)[44].

**Fig. 6 | EXO1 responds to formaldehyde-induced damage specifically. a** Left. IF staining against RPA2, RAD51 as well as cyclin A, and p53BP1 (pSer 1778) in RPE-1 WT and EXO1 KO (KO 7, KO 11) cells treated with 0 or 100 μM formaldehyde for 18 h. Right. Foci quantification performed in cyclin A or EdU positive cells. Data are shown with mean ± SEM from three independent experiments. ns: non-significant, $*p < 0.1$, $**p < 0.01$, and $****p < 0.0001$ (one-way ANOVA, followed by Kruskal–Wallis test). **b** Protein levels of EXO1, RPA32 (pSer4/Ser8) and RPA70 in RPE-1 WT and EXO1 KO (KO 7, KO 11) cells without treatment or treated with 100 μM formaldehyde for 18 h, with β-actin as a loading control. **c** Protein levels of EXO1 in RPE-1 WT cells with or without 1 mM formaldehyde treatment (30 min or 1 h), in either chromatin fraction (anti-histone H3) or non-chromatin fractions (α-tubulin). Quantification of the blots from three independent experiments is presented in Supplementary Fig. 6b. **d** Top. IF staining against pEXO1 (pSer 714) in S-phase determined by EdU staining in RPE-1 WT cells with or without 100 μM formaldehyde treatment for 18 h. Bottom. Foci quantification performed in EdU positive cells. Data are shown with mean ± SEM from three independent experiments. $****p < 0.0001$ (one-way ANOVA, followed by Kruskal–Wallis test).

TOP1cc accumulated in EXO KO cells, suggesting that it plays a rate-limiting role in TOP1cc repair (Fig. 7c, Supplementary Fig. 8a). Second, RADAR assays indicated that DPC accumulation, as measured by the smear density, is increased in EXO1 knockout cells compared to wild-type cells challenged with 100 μM formaldehyde (Supplementary Fig. 8b). Third, we performed an in vitro resection assay in the presence of dsDNA containing a DPC (Supplementary Fig. 8c–e). Briefly, we first linearized the pJLS2 plasmid crosslinked to DNA HpaII Methyltransferase (M.HpaII) protein[11] leading to linear dsDNA with a DPC located at 200 bp from the 5′ end. As EXO1 is a 5′ to 3′ exonuclease, we radiolabeled 3′ end extremities via a filling reaction with [α-$^{32}$P]-CTP. As a control, a linearized plasmid DNA without DPC was also radiolabeled. After digestion of both probes by AatII, we observed a shift in the migration profile of the DPC probe compared to non-DPC, showing the purity of the substrate (Supplementary Fig. 8d). Wild-type purified EXO1 (Fig. 7d) resected almost 100% (99.63%) of the non-DPC probe, as shown by the upper band disappearance (Fig. 7e, lane 2). As a specificity control, we used purified EXO1 D173A nuclease dead mutant[45] (Fig. 7d), which did not generate any significant resection (Fig. 7e, lane 3 and lane). We next challenged EXO1 with the DPC probe. Again, almost 100% (99.61%) of the probe was digested, but we observed a low proportion (7.1 and 4.7%) of the probe that was not fully degraded, leading to two by-products, products 1 and 2 (Fig. 7e lane 5). Even if EXO1 can go through DPC, as more than 80% of the substrate is fully resected, a DPC can slightly block DNA resection on the resected strand and the complementary strand (Supplementary Fig. 8c). We can conclude that EXO1 processes DPCs in vitro. Given that the major resectosome contains EXO1 but also MRE11-RAD50-NBS1 (MRN), BLM helicase and RPA[46,47], we investigated whether the whole resection machinery, MRN-RPA-BLM-EXO1, could process DPC in vitro using the same method. We observed that the whole resection machinery processed 91.3% of non-DPC and 83.2% of DPC probes (Supplementary Fig. 8e). Taken together, these data suggest that EXO1 can counteract the genotoxic effects of formaldehyde acting on both ICL and DPC lesions.

Our results suggest that EXO1 is implicated in ICLs/DPCs repair, which could support or affect the FA pathway. Consequently, using FANCD2 foci formation as a readout, we first investigated whether the FA pathway was functional after formaldehyde treatment in the EXO1 knockout context. EXO1 knockout led to an increase in FA pathway activation as monitored by FANCD2 foci formation (Fig. 8a) and western blotting (Supplementary Fig. 9a). To assess the epistatic status between the FA pathway and EXO1-mediated formaldehyde sensitivity, we performed an epistatic analysis in RPE2 cells knockout for FANCA. Knockdown of EXO1 in FANCA−/− cells led to additive formaldehyde sensitivity compared to FANCA−/− alone, suggesting that they work in parallel pathways leading to the repair of formaldehyde-induced lesions (Fig. 8b). The same pattern was observed in FANCA-deficient VU1365 cells[48] (Supplementary Fig. 9b). Consistent with the notion that EXO1 and the FA pathway function in parallel pathways, FANCA deficiency did not impair EXO1 recruitment to TMP-induced DNA lesions (Fig. 8c). We also investigated whether EXO1 shares some potential redundancy with other ICL repair nucleases, as FAN1 partially compensates for the lack of EXO1 in mismatch repair[49]. Co-depletion of FAN1 and FANCA, or FAN1 and EXO1, did not increase further

formaldehyde sensitivity suggesting that they are epistatic (Supplementary Fig. 9c).

## EXO1 depletion induces genomic instability and quadriradial chromosomes

We showed that EXO1 is important to counteract DNA replication stress and DNA damage induced by formaldehyde in S-phase, two phenomena that can lead to genomic instability. DNA replication stress can lead to DSBs in the next cell cycle where chromosomal fragile sites are shielded in large 53BP1 nuclear bodies in G1 phase[50,51]. To assess 53BP1 bodies as a surrogate for genome instability, we performed immunofluorescence against 53BP1 in G1 phase in cyclin A negative cells after formaldehyde treatment. Whereas the absence of EXO1 did not impact the number of 53BP1 bodies in untreated conditions, cells knockout for EXO1 showed a much higher number of 53BP1 bodies following formaldehyde treatment. Indeed, a significant proportion of cells presented more than ten 53BP1 bodies (Fig. 9).

Mutations in FA genes also lead to genome rearrangements and characteristic quadriradial chromosomes after ICL induction[52]. Consequently, we checked whether EXO1 inactivation could exacerbate quadriradial chromosome formation after MMC treatment. To do so, we used mouse embryonic fibroblasts (MEFs). First, similar to RPE-1 cells, Exo1 deficiency leads to formaldehyde sensitivity. Like in EXO1 KO RPE-1 cells, formaldehyde sensitivity in Exo1 KO MEFs was stronger than sensitivity to MMC, suggesting a more specific role in formaldehyde response than ICL-inducing agents (Fig. 10a, b). We then performed metaphase spreads after two days of MMC treatment at 90 nM (Fig. 10c), a dose that still allowed cells to grow (Fig. 10b). In untreated conditions, inactivation of Exo1 induced a slight but not significant increase in chromosomal instability, considering either breaks or quadriradial chromosomes. In WT MEFs, MMC treatment induced an increase in chromosome damage that was significant only for overall breaks but not for quadriradial chromosomes, which were more specific to ICL exposure (Fig. 10c). Conversely, under MMC treatment, inactivation of Exo1 had a much stronger effect on chromosome instability, not only in terms of breaks per metaphase but, most importantly, regarding quadriradial chromosomes (Fig. 10c). These results suggest a specific role for EXO1 in the generation of quadriradial chromosomes after ICL induction, a typical feature of FA cells. Another cellular phenotype of FA includes a marked cell-cycle delay with 4 N DNA content after the introduction of ICLs[53]. Similarly, EXO1 depletion in normal or formaldehyde conditions led to an increase in 4 N cells (Fig. 10d). Altogether, these results show that EXO1 is important in limiting genomic instability induced by formaldehyde treatment.

## Discussion

Formaldehyde is an abundant metabolic by-product with critical implications for the development of bone marrow failure in FA. Using a CRISPR-Cas9 screen, we have identified EXO1 as a regulator of formaldehyde-induced lesions. Several of our observations implicate EXO1 in ICL/DPC repair linked to the FA pathway. Loss of EXO1 leads to: (i) formaldehyde sensitivity as well as MMC sensitivity; (ii) the accumulation of TOP1cc DPC intermediates in cellulo; (iii) DNA replication stress as measured by fork progression and nascent strand

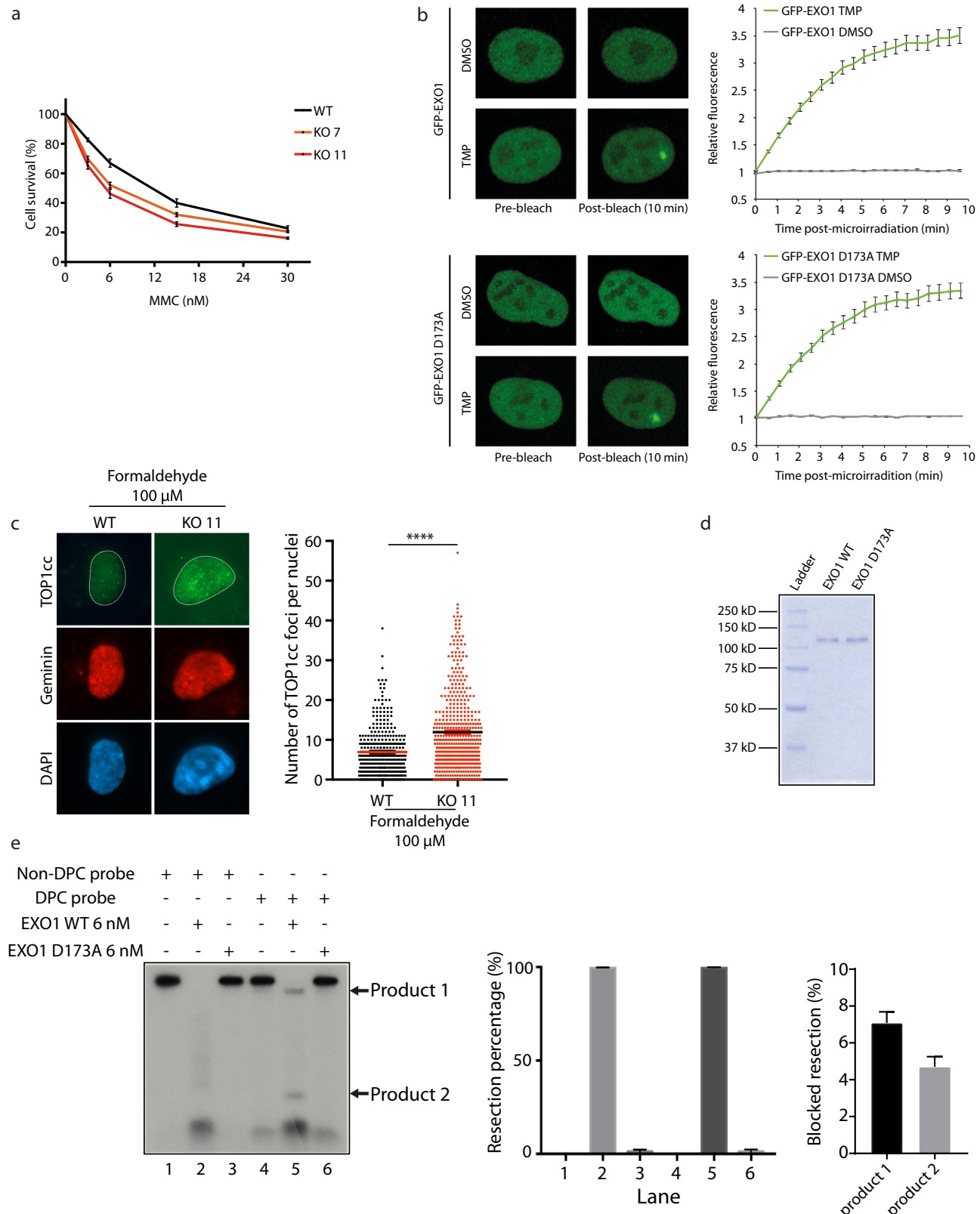

**Fig. 7 | EXO1 participates in repairing both ICLs and DPC. a** Survival curve of RPE-1 EXO1 WT, KO 7, KO 11 treated with different concentrations of MMC for 96 h. Data are presented with ±SEM from three independent experiments. **b** Quantitative evaluation of recruitment kinetics for GFP-EXO WT or GFP-EXO1 D173A to ICL sites induced by TMP incubation followed by UV micro-irradiation in HeLa cells. Mean curves ± SEM are shown ($n \geq 100$ cells per condition). **c** IF staining against TOP1cc in S/G2 phases marked by Geminin staining in RPE-1 WT and EXO1 KO 11 cells treated with 100 μM formaldehyde for 18 h. Data are shown with mean ± SEM from three independent experiments. ****$p < 0.0001$ (Mann–Whitney test). Quantification was performed in Geminin positive cells. **d** SDS-PAGE of purified human EXO1-WT and EXO1-D173A (nuclease dead mutant). **e** In vitro resection products of incubating purified EXO1-WT or EXO1-D173A with non-DPC or DPC probes were detected by autoradiography after agarose gel electrophoresis. The percentage of resection and the percentage of blocked resection (product 1 and product 2) were quantified.

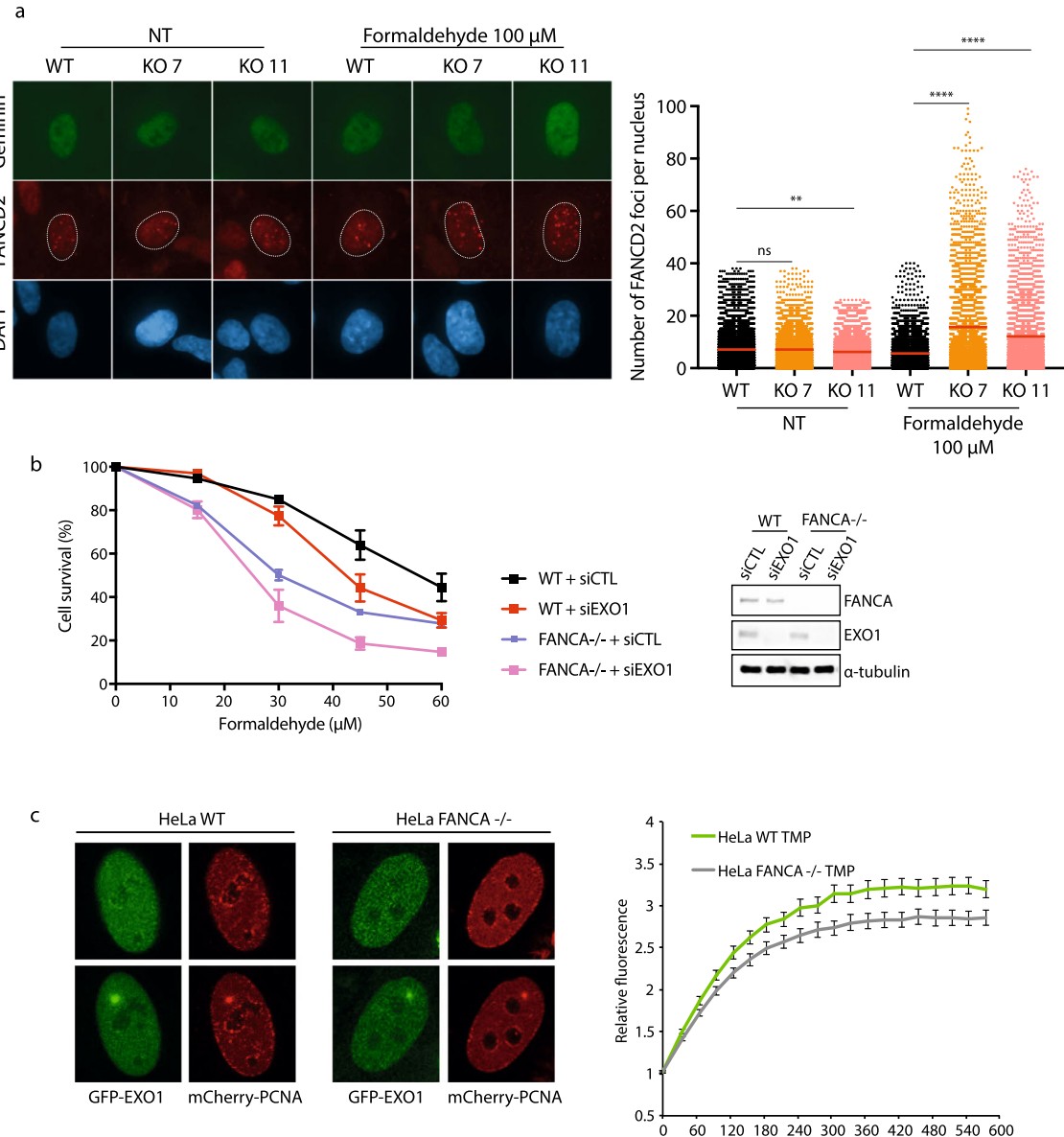

**Fig. 8 | EXO1 and the FA pathway contribute to formaldehyde resistance.**
**a** Right. IF staining against FANCD2 and Geminin was performed in RPE-1 WT, EXO1 KO (KO 7, KO 11) cells treated with or without 100 μM formaldehyde for 18 h. Data are shown with mean ± SEM from three independent experiments. ns: non-significant, **p < 0.01 and ****p < 0.0001 (one-way ANOVA, followed by Kruskal–Wallis test). Left. Quantification was performed in Geminin positive cells. **b** Left. Survival curve of RPE2 WT and FANCA KO cells transfected with siCTL or siEXO1 and treated with different concentrations of formaldehyde for 96 h. Data are presented with ±SEM from three independent experiments. Right. FANCA and EXO1 protein levels with α-tubulin as a loading control. **c** Left. Quantitative evaluation of recruitment kinetics for GFP-EXO1 to ICL sites induced by TMP incubation followed by UV micro-irradiation in S-phase, indicated by mCherry-PCNA staining, in HeLa WT and FANCA KO cells. Right. Mean curves ± SEM are shown (n ≥ 100 cells per condition).

degradation; (iv) an accumulation of quadriradial and 4 N chromosomes. All these data converge in a model whereby EXO1 is recruited to ICL/DPC lesions to prevent the accumulation of replicative stress and DSB formation. In the absence of EXO1, ICLs/DPCs are not properly repaired leading to DSBs. These DSBs are at least partially Mus81-dependent and can be processed by HR[17] and NHEJ[54].

Our data reinforce the concept that aldehydes cause a variety of DNA stresses, including DSB formation, replication stress, ICLs and DPCs, also renewing the debate about which DNA lesions cause FA[15]. Interestingly, we observed that formaldehyde induces DSBs specifically in S-phase, which is suggestive of DNA replication stress and known to activate the FA pathway. Based on our epistatic analysis showing that simultaneous depletion of EXO1 in FANCA causes additive formaldehyde sensitivity, we propose the following model where

EXO1 functions in parallel to the FA pathway at formaldehyde-stalled replication forks (Fig. 10e). Depletion of EXO1 leads to the accumulation of DPCs and DSBs in S-phase, activating the FA pathway, as measured by FANCD2 foci. However, the activation of the FA pathway alone is insufficient to compensate for the absence of EXO1, leading to decreased survival after formaldehyde treatment. Unrepaired lesions will eventually form DSBs that will be repaired by canonical HR, which relies on the downstream function of EXO1 in DNA resection. In this DNA resection step, the nuclease activity of DNA2 can normally compensate EXO1 deficiency after irradiation. However, DNA2 did not appear in the top hits of our CRISPR-Cas screen, suggesting that EXO1 acts independently to resolve formaldehyde-induced damage.

EXO1 is a 5′ to 3′ double-stranded DNA exonuclease with a modest endonuclease or 5′ flap activity[55]. Its strong capacity for degrading

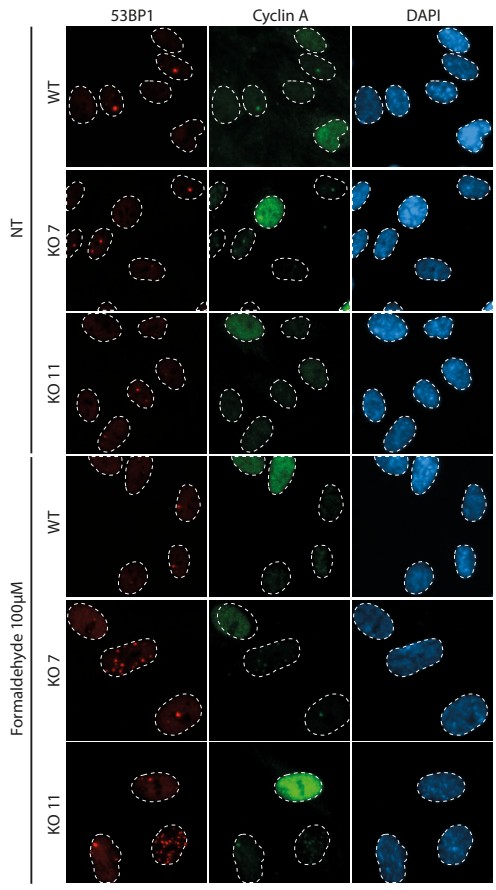

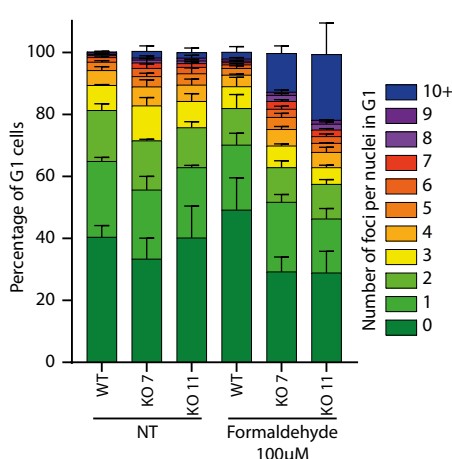

**Fig. 9 | EXO1 deficiency leads to 53BP1 bodies accumulation.** Quantification of immunofluorescence analysis against 53BP1 nuclear bodies in G1 (cyclin A negative cells) in RPE-1 WT, EXO1 KO7, or EXO1 KO11 cells, untreated or treated 100 µM formaldehyde for 18 h. The histogram presents the percentage of G1 cells corresponding to different number of 53BP1 bodies. Data are shown with mean ± SEM from three independent experiments.

double-stranded DNA must be restrained in some conditions. EXO1 can be detrimental to the cell as it contributes to fork degradation in BRCA1- and BRCA2-deficient cells upon HU treatment[56]. In wild-type cells, our experiments highlight a beneficial role for EXO1 as it limits DNA replication stress and fork degradation induced by formaldehyde. Such a protective effect of EXO1 loss on DNA replication forks was reported previously after HU and CPT treatment[36]. EXO1-depleted cells also elicited elevated γH2AX staining and the accumulation of DSBs in S-phase. These formaldehyde-induced DSBs led to the activation of Chk1 and were dependent on the structure-specific nuclease Mus81, which acts at regressed forks and can lead to chromosomal instability. Of note, it was reported that Mus81 depletion did not alter the sensitivity of mouse Exo1 null cells to low-dose CPT treatment[36]. Since a rescue of γH2AX accumulation in EXO1 KO cells is observed with formaldehyde treatment, it indicates some degree of specificity for the damage type. Globally, we envision that EXO1 removes ICLs/DPCs that impede DNA replication.

Using an unbiased proteomic approach to study protein recruitment to psoralen-ICLs, it was previously shown that EXO1 is associated with cross-linked chromatin[57]. Herein, we show that EXO1 is recruited to chromatin damaged by formaldehyde and TMP-induced ICLs but not angelicin-mediated monoadducts, suggesting EXO1 responds to DPCs/ICLs rather than monoadducts. We also noticed that the recruitment of EXO1 to TMP-induced ICLs also occurred outside of S-phase. This is consistent with an MMR-dependent ICL repair mechanism, which requires the nuclease activities of MutLα and EXO1[58].

At the structural level, EXO1 protein has several features that serve very well in DPC removal. EXO1 has no DNA sequence specificity and,

therefore, would accommodate the processing of the heterogeneous nature of crosslinked proteins. It also bears a DNA nuclease domain that could remove DPCs. In each case, a gap would be left for a polymerase to fill in. We did observe different resection patterns when we used EXO1 alone versus the MRN-RPA-BLM-EXO1 resection machinery. MRN-RPA-BLM-EXO1 did not produce two minor by-products, possibly due to BLM helicase activity unwinding the DNA. Thus, mammalian cells have evolved distinct regulatory systems for exo/endonucleolytic DPC degradation. Firstly, an EXO1-dependent mechanism influencing FA pathway activation after formaldehyde treatment, and secondly, a MRN-CtIP mechanism which removes covalently linked Top2 and Top1 following etoposide and camptothecin treatment, respectively[14,59,60]. Consistent with this, our data support the prediction that EXO1 deficient cells would be sensitive to camptothecin and etoposide, which was indeed reported previously[17,61].

Remarkably, our study with Exo1−/− MEFs has shown an increase in quadriradial chromosomes, a hallmark of FA cells. This is consistent with a previously described mouse model inactivated for Exo1[62] showing phenotypes similar to FA mouse models[16]. Not only did Exo1−/− male and female mice show meiosis defects and sterility, but male mice also showed a higher rate of spontaneous quadriradial chromosomes in spermatocytes than the control[62]. Although unexplained at the time, these phenotypes are characteristic of FA mouse models. The generation of double knockout mice of Exo1 and a FA gene would be valuable to investigate the consequence of EXO1 deletion on the progression of the disease. Furthermore, it will be paramount in the future to assess whether EXO1 can be identified as a new FA gene in unassigned FA patients or if there are characterized FA patients

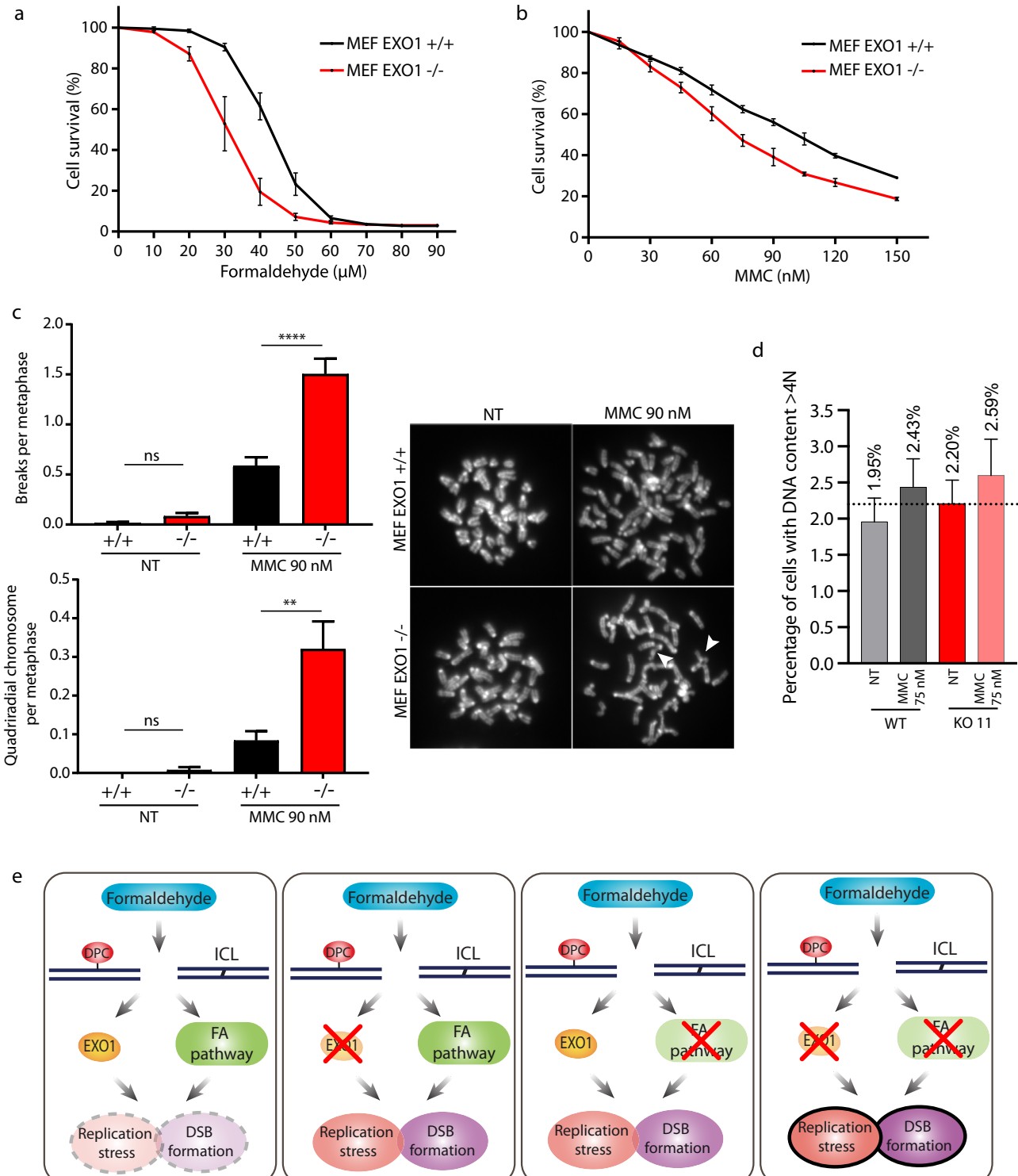

**Fig. 10 | EXO1 deficiency leads to heightened chromosomal breaks, quadriradial chromosomes, and accumulation of 4 N DNA. a** Survival curve of mouse embryonic fibroblasts (MEFs) Exo1 WT and Exo1 KO treated with different concentrations of formaldehyde for 96 h. Data are presented with ±SEM from three independent experiments. **b** Survival curve of MEFs Exo1 WT and EXO1 KO treated with different concentrations of MMC for 96 h. Data are presented with ±SEM from three independent experiments. **c** Quantification of chromosomal breaks and quadriradial chromosomes per metaphase in MEFs Exo1 WT and Exo1 KO MEFs under 0 or 90 nM MMC for 48 h. **$p < 0.01$ and ****$p < 0.0001$ (one-way ANOVA,

followed by Kruskal–Wallis test). **d** Percentage of cells with DNA content above 4 N in RPE-1 WT and EXO1 KO 11 cells after 24 h of 75 nM MMC treatment. **e** A potential model of how EXO1 participates in response to formaldehyde-induced DNA damage. Formaldehyde induces ICLs/DPCs, which are acted upon by the EXO1 and the FA pathway. When EXO1 or FA pathways are inactivated, this leads to aggravated DNA replication stress and DSB formation. The EXO1 or FA pathways cannot compensate totally for each other as they show additive functions in epistatic analyses.

bearing EXO1 mutations as a driver of the disease. In this case, these patients might be susceptible to endogenous formaldehyde levels, and their clinical outcomes should be closely monitored.

## Methods

### Cell lines, antibodies, and reagents

RPE-1 hTeRT (RPE-1) cells were a gift from Dr. Graham Dellaire (Dalhousie University) and maintained in Dulbecco's modified Eagle's medium (DMEM) supplemented with 10% fetal bovine serum (FBS) (Hyclone-Thermo Fisher Scientific, Ottawa, Canada). RPE-1 hTeRT cells stably expressing Cas9 (RPE-1-Cas9) were obtained from Dr. Dan Durocher (The Lunenfeld-Tanenbaum Research Institute, Toronto) and maintained in DMEM supplemented with 10% FBS. Mouse embryonic fibroblasts (MEFs, p53−/−) proficient for EXO1 (MEF WT) or deficient for EXO1 (MEF KO) were generously provided by Dr. Winfried Edelmann (Albert Einstein College of Medicine) and maintained in DMEM supplemented with 10% FBS. RPE2 and VU1365 were gifted by Dr. Josephine Dorsman (Amsterdam UMC). RPE2 cells were maintained in DMEM/F-12 supplemented with 10% FBS and 15 µg/ml hygromycin. VU1365 cells were maintained in DMEM supplemented with 10% FBS and 300 µg/ml G418. HeLa FANCA knockout (HeLa FANCA KO) cells were received from Dr. Alan D'Andrea (Dana-Farber Cancer Institute) and cultured in DMEM supplemented with 10% FBS. RPE-1 EXO1 knockout (EXO1 KO) cells were generated separately by IDT and Synthego strategies, and RPE-1 EXO1 complemented cells were generated via the AAVS1 system and maintained in DMEM supplemented with 10% FBS[31]. HeLa cells were authenticated using Short Tandem Repeat (STR) analysis by ATCC and maintained in DMEM supplemented with 10% FBS. All cell lines were grown at 37 °C, 5% $CO_2$, and routinely tested to be mycoplasma free.

The list of antibodies and reagents and resources used in this study are listed in Supplementary Tables 1 and 2, respectively.

### CRISPR-Cas 9 screen

RPE-1 hTERT cells stably expressing Cas9 were infected with TKO_v1 CRISPR library[25] (Addgene) at a MOI (Multiplicity Of Infection) of 0.3 in DMEM supplemented with 10% FBS, 10 mM HEPES (Life Technologies) and 8 µg/ml polybrene (Millipore-Sigma). After 18 h, cells were washed once with PBS and put in fresh media for 24 h. Infected cells were then selected with 25 µg/ml puromycin for 48 h. A plate without lentivirus was kept in parallel to check that the puromycin selection killed all uninfected cells. A second plate with lentivirus without any selection was used to calculate the MOI. After the release from puromycin, cells were kept in culture for 6 days. Infected cells were then divided into two conditions, either untreated or treated with formaldehyde at 70 µM, a dose that kills ~80% of the cells in 15 days (Supplementary Fig. 1a). Cells were passed every 3 days into fresh formaldehyde-containing media, maintaining a 400-fold representation of the sgRNA library. The screen was performed in two replicates. Cells were collected before and after 9 and 15 days of treatment for both the treated and untreated conditions. Genomic DNA was then extracted with the QiAamp Blood Maxi Kit (Qiagen), adding $15 \times 10^6$ cells per column according to the manufacturer's instructions. Elution was performed using 600 µl of Buffer AE and by reloading the eluate on the column with an extra 200 µl of Buffer AE. Genomic DNA precipitation was performed by adding NaCl to a final concentration of 0.2 M and adding 2 volumes of −20 °C EtOH 100%. After centrifugation, DNA pellets were washed once with 70% EtOH and air-dried for 5–10 min. The DNA pellets were resuspended by repeated pipetting in 400 µl of 10 mM Tris−HCl pH 7.5, heated at 55 °C for 1 h and well resuspended by quick vortex and pipetting up and down 10 times. Samples were quantified with Nanodrop, and the concentration was adjusted to 500 ng/µl. Then, the first round of PCR was performed on an amount of genomic DNA corresponding to 400-fold the library. Since the genome of 1 cell contains ~6.5 pg of DNA[63], PCR reactions were performed on 240 µg of

DNA. For each reaction, 5 µg of gDNA was mixed with 25 µl of 2X KAPA HiFi Master Mix (KAPA HiFi HS RM from Roche), with primers at 1 µM final concentration (Fwd: AGGGCCTATTTCCCATGATTCCTT, Rev: TCAAAAAAGCACCGACTCGG) and water to a final volume of 50 µl. The PCR was performed using the following program: 3 min at 95 °C, 30 s at 98 °C, 30 s at 60 °C, and 30 s at 72 °C. The last three steps were repeated 19 times, and then samples were maintained at 72 °C for 1 min and cooled down to 4 °C. For each condition, PCRs were pooled together, and 10 µL was loaded on a 2% agarose gel to check the presence of the expected ~350 bp product. Then, a second PCR was performed to identify sgRNA from each condition with Illumina TruSeq adapters with i5 and i7 indexes. For each sample, 2 PCRs were performed. Samples were prepared as follows: 2.5 µl of PCR1 product, 25 µl of 2X KAPA HiFi Master Mix, 5 µl of each Illumina primers at 10 µM, and water to 50 µl. The PCR conditions were as follows: 3 min at 95 °C, 30 s at 98 °C, 30 s at 65 °C, 30 s at 72 °C. The last three steps were repeated 17 times, and then the samples were maintained at 72 °C for 1 min and cooled down to 4 °C. For one condition, the two reactions were pooled together and purified using the QIAquick PCR purification kit (Qiagen). Elution products were migrated on a 2% agarose gel. The 200 bp PCR products were purified using the QIAquick gel extraction kit (Qiagen). Eluates were purified again with the QIAquick PCR purification kit and sent to sequencing on an Illumina HiSeq 4000, standard single read, 50 bp.

Raw sequencing result data files (i.e., FASTQ files) quality was assessed by FastQC analysis. MAGeCK count command[64] was used to identify library adapter length from FASTQ files, perform primary alignment against the TKO_v1 library, and provide sgRNA read counts. Once adapter lengths were determined, resulting sequences were trimmed and re-aligned against the TKO_v1 library with standard parameters on Bowtie2[65]. Read count files for each condition were then re-generated with MAGeCK's (v0.5.9.3) count command using Bowtie-aligned sequences. Variation in sgRNA read counts after treatment, and individual gene level NormZ scores (Normalized Z-scores of up to 6 sgRNAs per gene) were obtained with the DrugZ algorithm (v1.1.0.2), a program developed for CRISPR screen analysis, running with default parameters (default parameters include: paired samples and a pseudocount of 5 added to each sgRNA count)[26]. NormZ scores for each time point are represented in Supplementary figure 1b and available in Supplementary Data 1. Screen quality was assessed by estimating core essential genes dropout rate over time compared to core non-essential genes, as seen in ref. [66].

To select potential hits, we first determined the list of genes with a NormZ lower than −3, representing sensitivity hits at a probability ≤ 0.001 in both Day 9 and Day 15 conditions and then applied an FDR threshold at 0.05 to consider a gene as significantly decreased.

### siRNA screening

Plates containing ON-TARGETplus SMARTpool siRNAs candidates from the CRISPR-Cas9 screen were purchased from Dharmacon. For each of these candidates, RPE-1 cells were seeded at a density of 2000 cells per well in Corning 3603 black-sided clear bottom 96-well plates and reverse transfected with 50 nM siRNA using Lipofectamine RNAiMAX transfection reagent (Invitrogen life technology). The next day, cells were exposed to 40 µM formaldehyde (J.T.Baker) diluted in culture medium, the concentration at which the most significant difference between the siCTL and the siADH5 or siFANCA was seen. (Supplementary Fig. 2a). After 96 h of treatment, nuclei were stained with Hoechst 33342 (Invitrogen life technology) at 10 µg/ml in medium for 30 min at 37 °C. Images of entire wells were acquired at 4x with a Cytation 5 Cell Imaging Multi-Mode Reader (Biotek) followed by quantification of Hoechst-stained nuclei with the Gen5 Data Analysis Software V3.03 (BioTek Instruments). Cell survival was expressed as a fold change of survival in treated cells relative to vehicle (medium)-treated cells. Results represent the

mean ± SEM of three independent experiments performed in technical triplicate.

## Protein extraction and Immunoblotting (Western Blot)

Cells were collected and lysed in lysis buffer containing 300 mM NaCl, 1% Triton X-100, 50 mM Tris-HCl pH8, 5 mM EDTA, and 1 mM DTT supplemented with protease inhibitors (1 mM PMSF, 3.4 µg/ml Aprotinin and 1 µg/ml Leupeptin) and phosphatase inhibitors (5 mM NaF and 1 mM $Na_3VO_4$) for 30 min on ice. Samples were sonicated using a Bioruptor sonicator (Diagenode) for 10 cycles (30 s ON/OFF at high power) and centrifuged for 20 min at 4 °C. Supernatants were collected and dosed by Bio-Rad Protein Assay Dye Reagent. Equal amounts of total protein were separated by SDS–PAGE and then transferred to nitrocellulose membrane (BioRad) and immunoblotted with antibodies.

## Generation of RPE-1 EXO1 KO cell lines by CRISPR-Cas9

**IDT strategy.** Alt-R CRISPR-Cas9 EXO1 crRNA targeting the genomic sequence AAGCTCGAGAGTGTTTCACC (ID: Hs.Cas9.EXO1.1.AC, IDT) and Alt-R CRISPR-Cas9 tracrRNA were duplexed and transfected along with the Alt-R S.p. HIFI Cas9 nuclease V3 into RPE-1 cells by RNAiMAX lipofection following IDT instructions. Clones were then isolated, and knock-out was confirmed by immunoblotting against EXO1 and Sanger sequencing following genomic DNA extraction with the Qiagen QIAMP DNA mini and Blood mini kit. PCR amplification of genomic DNA was performed using primers GCCTAAAGCATCTGGGTTAATG (chr1:241853193-241853214, GRCh38/hg38 Assembly) and TGTTCCCTTCTCCTTCTGACAT (chr1: 241853602-241853623, GRCh38/hg38 Assembly). One KO clone, designed as KO 7, was selected for the following experiments.

**Synthego strategy.** The multi-guide sgRNA targeting EXO1 (guide 1: uuuggcaccauggggauaca, guide 2: ugugaggaaguauaaagggc and guide 3: ccuaucacguagguucaccuu) and the S.p. Cas9 2NLS nuclease were obtained with Synthego's Gene Knockout kit v2 and transfected into RPE-1 cells using RNAiMAX lipofection following Synthego instructions. Genomic DNA was extracted from the transfected pool of cells using Qiagen QIAMP DNA mini and Blood mini kit, and PCR amplified using primers AACCAATTTCTGCATTGGACTC (chr1:241850300-241850321, GRCh38/hg38 Assembly) and GAAGGAAGTCATCCCTGATTTG (chr1:241850727-241850748, GRCh38/hg38 Assembly). Editing efficiency in the transfected cell pool was determined by amplicon Sanger sequencing followed by online inference of CRISPR Edits (ICE) analysis (https://www.synthego.com/products/bioinformatics/crispr-analysis). Clones were then isolated and tested both by immunoblotting against EXO1 and DNA sequencing. One selected clone for EXO1 knock-out was named KO11 and used thereafter.

## Generation of RPE-1 EXO1 complemented cell lines

EXO1 sequence flanked with NotI and BstBI restriction enzyme sites was inserted into AAVS1 neo[R] vector (kindly provided by Dr. Jacques Côté/Yannick Doyon, Université Laval). RPE-1 WT cells were seeded in a six-well plate with a density of 300000 cells per well and transfected with AAVS1 neo[R] vector, referred as empty vector (EV), while KO11 cells were transfected with either AAVS1 neo[R] vector or EXO1- AAVS1 neo[R] vector, using Lipofectamine 2000 transfection reagent (Invitrogen life technology) with plasmids encoding Zinc-Finger nucleases[31]. G418 sulfate (Wisent INC/Multicell) treatment was used to select transfected cells 24 h after transfection. Clones were then generated from the pool of transfected cells and tested by immunoblotting against EXO1.

## Survival assays

For survival assays in RPE-1 WT, RPE2, and VU1365 cells, 150,000 cells were seeded into one well of a six-well plate for 18 h and then transfected with 20 nM siCTL (UUCGAACGUGUCACGUCAA), siEXO1 (CAAGCCUAUUCUCGUAUUU), siDNA2 (AUAGCCAGUAGUAUUCGAU), siBRCA2 (GAAGAAUGCAGGUUUAAUAUUdTdT), or siFAN1 (GCAGGAGAAGGGAAUUGUAACUAAA) using Lipofectamine RNAiMAX transfection reagent (Invitrogen life technology). Cells were then seeded in triplicates into Corning 3603 black-sided clear bottom 96-well microplates at a density of 2000 cells per well 24 h after transfection. Pellets of the remaining cells were kept and stored at −80 °C until processed for protein extraction and immunoblotting. For survival assay in RPE-1 WT, KO 7, KO 11, and RPE-1 EXO1 complemented cell lines, cells were seeded in triplicates into Corning 3603 black-sided clear bottom 96-well microplates at a density of 2000 cells per well. For MEFs, cells were seeded at a density of 1200 cells per well. Once attached to the plate, cells were exposed to different concentrations of formaldehyde (J.T.Baker), or Mitomycin C (Sigma) diluted in culture medium. After 96 h of treatment, nuclei were stained with Hoechst 33342 (Invitrogen life technology) at 10 µg/ml in medium for 30 min at 37 °C. Images of entire wells were acquired at 4x with a Cytation 5 Cell Imaging Multi-Mode Reader followed by quantification of Hoechst-stained nuclei with the Gen5 Data Analysis Software V3.03 (BioTek Instruments). Cell survival was expressed as a percentage of survival in treated cells relative to vehicle (medium)-treated cells. Results represent the mean ± SEM of at least three independent experiments performed in technical triplicate. Additional statistical analyses can be found in Supplementary Data 2.

## Immunofluorescence staining

**γH2AX and p53BP1 (S1778) foci.** For the detection of γH2AX and p53BP1 (S1778) foci formation, cells were incorporated with 10 µM EdU for 20 min at 37 °C and pre-extracted with NuEx buffer (20 mM HEPES pH 7.4, 20 mM NaCl, 5 mM $MgCl_2$, 0.5% NP40, 1 mM PMSF, 3.4 µg/ml Aprotinine, 1 µg/ml Leupeptin, 5 mM NaF, 1 mM $Na_3VO_4$ and 1 mM DTT) for 10 min on ice. This method removes nucleoplasmic signals and helps in the detection of foci. Cells were washed twice with phosphate-buffered saline (PBS) followed by fixation with 4% paraformaldehyde (w/v) in PBS for 10 min at room temperature. After three washes with PBS, cells were permeabilized in 0.5% TritonX-100 in PBS for 10 min at room temperature and blocked in 2% bovine serum albumin (BSA) in PBS for 30 min at room temperature. Click-iT™ EdU Cell Proliferation Kit for Imaging, Alexa Fluor™ 647 dye (Thermo Fisher) was used to label EdU in cells to define cell cycle status by applying the Click-iT reaction cocktail for 30 min at room temperature. Cells were washed once with PBS and incubated with primary antibodies against γH2AX (1:5000, EMD Millipore) and pS1778-p53BP1 (1:400, Cell signaling technology) for 90 min at room temperature. After three washes with PBS, cells were stained with Alexa Fluor 568 goat anti-rabbit (1:1000, Thermo Fisher) and Alexa Fluor 488 goat anti-mouse (1:1000, Thermo Fisher) for 1 h at room temperature. Nuclei were stained for 10 min with 1 µg/ml 4,6-diamidino-2-phenylindole (DAPI) before mounting onto slides with ProLong® Gold Antifade Mountant (Invitrogen life technology).

**RAD51, FANCD2, and 53BP1 bodies.** For RAD51 and 53BP1 bodies immunodetection, unless otherwise stated, all dilutions were prepared in PBS and incubations were performed at room temperature with intervening washes in PBS. Cell fixation was carried out by incubation with 4% paraformaldehyde for 10 min followed by prechilled methanol for 5 min at −20 °C. This was followed by permeabilization in 0.2% TritonX-100 for 5 min and a quenching step using 0.1% sodium borohydride for 5 min. After blocking for 1 h in a solution containing 10% goat serum and 1% BSA, cells were incubated for 1 h with primary antibodies anti-RAD51 (1:5000, B-bridge International) and anti-cyclin A (1:400, BD Biosciences), anti-FANCD2 (1:1000, Novus) and anti-Geminin (1:2000, Abcam) or 53BP1 (1:1000, Novus) and anti-cyclin A diluted in 1% BSA.

Secondary antibodies Alexa Fluor 568 goat anti-rabbit (1:1000, Thermo Fisher) and Alexa Fluor 488 goat anti-mouse (1:1000, Thermo Fisher) were diluted in 1% BSA and incubated for 1 h. Coverslips were mounted onto slides with ProLong® Gold Antifade Mountant with DAPI (Invitrogen life technology).

**RPA foci.** Cells were pre-extracted with RPA buffer (25 mM HEPES pH 7.9, 300 mM sucrose, 50 mM NaCl, 1 mM EDTA, 3 mM $MgCl_2$, and 0.5% Triton X-100) for 5 min on ice twice, followed by fixation with 4% paraformaldehyde for 20 min at room temperature. After two PBS washes, permeabilization was carried out in 0.5% TritonX-100 for 15 min. After one wash with PBS supplemented with 0.1% tween 20 (0.1% PBST), 2% BSA-PBS was used for blocking for 45 min. Cells were incubated with primary antibody against RPA34-19 (1:400, Calbiochem) for 2 h at room temperature, followed by three washes in PBST and incubation with Alexa Fluor 488 goat anti-mouse (1:1000, Thermo Fisher) secondary antibody for 1 h at room temperature. Coverslips were mounted onto slides with ProLong® Gold Antifade Mountant with DAPI (Invitrogen life technology).

**pEXO1 (S714) foci.** Cells were incorporated with 10 μM EdU for 20 min at 37 °C and washed twice with cold PBS before pre-extraction with RPA buffer as above for 3 min on ice twice. Cell fixation was carried out with 4% paraformaldehyde for 20 min at room temperature, followed by three washes with PBS and permeabilization with 0.5% PBST for 5 min. Cells were washed three times in PBS and incubated with primary antibody against pS714-pEXO1 (1:200, kind gift from Dr. Kum Kum Khanna) for 1 h at room temperature, succeeded by a wash in 0.1% PBST and three washes with PBS. Cells were then stained with Alexa Fluor 488 goat anti-mouse (1:1000, Thermo Fisher) for 1 h at room temperature. The coverslips were washed once with 0.1% PBST, followed by three washes with PBS. A second permeabilization was carried out by incubation with 0.5% TritonX-100 for 20 min, followed by a wash in PBS containing 3% BSA. Click-iT™ EdU Cell Proliferation Kit for Imaging, Alexa Fluor™ 647 dye (Thermo Fisher) was used to label EdU in cells to define cell cycle by applying the Click-iT reaction cocktail for 30 min at room temperature. Cells were washed once again with PBS containing 3% BSA. Nuclei were stained for 10 min with 1 μg/ml DAPI before mounting onto slides with ProLong® Gold Antifade Mountant (Invitrogen life technology).

**TOP1cc foci**[44]. For Topoisomerase I-DNA covalent complexes (TOP1cc) immunodetection, cells were fixed with 4% paraformaldehyde in PBS for 15 min on ice, followed by permeabilization in 0.25% TritonX-100 for 15 min at 4 °C. Subsequently, cells were incubated in 1% SDS at room temperature for 5 min followed by five washes in buffer containing 0.1% BSA and 0.1% TritonX-100 in PBS. Coverslips were incubated in blocking buffer containing 10% milk, 150 mM NaCl, and 10 mM Tris-HCl pH 7.4 for 1 h, followed by the addition of anti-Topoisomerase I-DNA Covalent Complexes Antibody, clone 1.1 A (1:250, Millipore-Sigma) and anti-Geminin (1:500, Proteintech) in 5% goat serum in PBS at 4 °C overnight. Cells were rinsed with washing buffer five more times. Secondary antibodies Alexa Fluor 568 goat anti-rabbit (1:1000, Thermo Fisher) and Alexa Fluor 488 goat anti-mouse (1:1000, Thermo Fisher were diluted in 5% goat serum in PBS and applied for 1 h. Cells were rewashed five times in washing buffer, nuclei were stained for 10 min with 1 μg/ml DAPI prior to mounting onto slides with ProLong® Gold Antifade Mountant (Invitrogen life technology).

**pDNAPKcs (S2056) foci.** For pDNAPKcs (S2056) immunodetection, cells were incorporated with EdU for 20 min at 37 °C. Unless otherwise stated, all immunofluorescence dilutions were prepared in PBS and incubations were performed at room temperature with intervening washes in PBS. Cell fixation was carried out

by incubation with 4% paraformaldehyde for 20 min followed by permeabilization in 0.2% TritonX-100 for 20 min. After blocking for 2 h in 10% FBS, Click-iT™ EdU Cell Proliferation Kit for Imaging, Alexa Fluor™ 647 dye (Thermo Fisher) was used to label EdU in cells to define cell cycle status by applying the Click-iT reaction cocktail for 30 min at room temperature. Cells were incubated for 1 h with primary antibody anti-pS2056-pDNAPKcs (1:1000, Abcam) and with secondary antibody Alexa Fluor 568 goat anti-rabbit (1:1000, Thermo Fisher) for 1 h. Nuclei were stained for 10 min with 1 μg/ml DAPI prior to mounting onto slides with ProLong® Gold Antifade Mountant (Invitrogen life technology).

## Cell fractionation
Cells from each indicated condition were collected by trypsinization, centrifugation at 1200 rpm for 4 min, and washed once in 1 ml PBS. Then, 250 μl of ice-old CSK buffer (300 mM sucrose, 200 mM NaCl, 3 mM $MgCl_2$, 0.5% Triton X-100, 1 mM EGTA, 10 mM PIPES and adjusted pH to 6.8) was added to each cell pellet on ice for 10 min. Subcellular fractions were centrifuged at 3000 g for 30 s at 4 °C and the supernatants were transferred to 1.5 ml tubes as the non-chromatin fraction. The cell pellets were then rinsed twice with 250 μl and twice with 500 μl CSK buffer by centrifuging at 3000 g for 30 s at 4 °C. After discard washes, 125 μl SDS buffer that contains 2% SDS, 50 mM Tris-HCl pH 7.4, and 10 mM EDTA was applied to each cell pellet, followed by several up and down pipetting directly. Samples containing chromatin fraction were heated at 95 °C for 2 min subsequent to adding 2X SDS loading buffer, sonicated using a Bioruptor sonicator (Diagenode) for 10 cycles (30 s ON/OFF at high power), and heated again at 95 °C for 7 min. 20 μl samples were loaded on 4–12% Bis-Tris gel (Life Technologies).

## Laser micro-irradiation
Live-cell microscopy and laser micro-irradiation were carried out with a Leica TCS SP5 II confocal microscope driven by Leica LAS AF software using a 63×/1.4 oil immersion objective. The microscope was equipped with an environmental chamber set to 37 °C and 5% $CO_2$. Briefly, HeLa cells seeded onto 35-mm fluorodishes (World Precision Instruments, Inc.) were transfected separately with 2 μg pEGFP-EXO1, 2 μg pEGFP-EXO1-D173A, 5 μg pOZ-mCherry-UHRF1 (received from Dr. Martin Cohn) or 2 μg pDEST-mCherry-LacR-NLS-CSB (received from Dr. Xu-Dong Zhu), using Lipofectamine 2000 transfection reagent (Invitrogen life technology). The next day, cells were treated with 4,5′,8-trimethylpsoralen (TMP) or angelicin (Sigma) at 20 μg/mL in the media for 30 min at 37 °C before being micro-irradiated in the nucleus for 200 ms using a 405 nm UV-laser (15% and 10% intensity for TMP and angelicin, respectively) at the following settings: format 512 × 512 pixels, scan speed 100 Hz, mode bidirectional, zoom 2×. To monitor the recruitment of proteins to laser-induced DNA damage sites, micro-irradiated cells were imaged every 30 s for 10 min or 20 min as indicated, after which fluorescence intensity of proteins at DNA damage sites relative to an unirradiated nuclear area was quantified and plotted over time. Kinetic curves represent the mean relative fluorescence intensity, and error bars show the SEM. For recruitment studies in S-phase HeLa WT and FANCA KO cells, 2 μg pEGFP-EXO1 was cotransfected with 1 μg mCherry-PCNA and mean kinetics curves were from cells exhibiting mCherry-PCNA-positive replication foci only. All results are from at least three independent experiments (total $n \geq 100$ cells).

## DNA combing
For Fig. 3b, RPE-1 cells were treated with culture medium or 100 μM formaldehyde for 18 h. Then, RPE-1 cells were incorporated with 50 μM IdU for 30 min, washed with prewarmed culture medium, and subjected to a 100 μM CldU pulse for 30 min. Both IdU and CldU pulse were performed in the presence or absence of

100 μM formaldehyde. At the end of the CldU pulse, cells were collected by trypsinization, washed once in cold PBS and resuspended in cold PBS at $1 \times 10^6$ cells/ml. For Fig. 3c, RPE-1 cells were incorporated with 50 μM IdU for 30 min, washed with prewarmed culture medium, and subjected to a 100 μM CldU pulse for 30 min, followed by a 4 h treatment of 100 μM formaldehyde or 2 mM HU. At the end of the treatment, cells were collected by trypsinization, washed once in cold PBS and resuspended in cold PBS at $1 \times 10^6$ cells/ml. Cells were incubated for 10 s at 42 °C, carefully mixed with an equal volume of 1% LMP agarose in PBS (Wisent Bio Products) and immediately transferred into a plug cast. Plugs were kept at room temperature for 25 min and then incubated at 4 °C in a humid chamber for 10 min. When polymerized, plugs were pushed in a 13 ml round bottom tube containing 500 μl/plug of proteinase K buffer (10 mM Tris-HCl pH7.5, 50 mM EDTA, 1% Sarkosyl and 2 mg/ml proteinase K). Plugs were then incubated for 48 h at 50 °C, cooled down to room temperature and washed five times/5 min in 5 ml of $TE_{50}$ (10 mM Tris-HCl, pH7.0, 50 mM EDTA) and kept at 4 °C until use. After five other washes of 5 min in 10 ml of TE (10 mM Tris-HCl, pH7.0, 1 mM EDTA), plugs were washed once for 5 min with MES buffer 1X (freshly prepared from MES buffer 10X, pH 5.8 (500 mM hydrate and 500 mM MES sodium salt were mixed according to the ratio of 70:30, and pH adjusted to 5.8 with MES sodium salt) diluted in $H_2O$). Each plug was then resuspended in 3 ml of 1X MES buffer and melted for 15 min at 65 °C. From this step onwards, DNA was in solution and tubes needed to be handled with extreme care. DNA solution was incubated 10 min at 42 °C and 3 units of agarase (New England Biology) were added for overnight incubation at 42 °C. DNA solution was then transferred into a Teflon reservoir (Genomic Vision), and DNA combing was performed on silanized coverslips (Genomic vision) at 300 μm/s using a DNA combing system (Genomic Vision). Coverslips were then incubated 2 h at 60 °C and fixed on a slide with cyanoacrylate glue. They were then successively incubated in 70%, 95% and finally 100% EtOH in Coplin Jars for 5 min and let to dry. DNA was then denaturized by a 22 min incubation in 1 M NaOH at room temperature and washed five times in PBS pH 7.4 for 1 min. Slides were then saturated in blocking buffer (PBS, 0.1% Triton X-100, 1% BSA) for 15 min at room temperature and incubated with two primary antibodies, anti-BrdU that recognizes IdU (1:20, Beckton Dickinson, mouse) and anti-BrdU that recognizes CldU (1:20, Abcam, rat) for 45 min at 37 °C in a humid chamber. After three 5 min washes in PBS-Triton (PBS containing 0.1% Triton X-100), slides were incubated with secondary antibodies anti-mouse IgG1 Alexa 488 and anti-rat Alexa 568 (Thermo Fisher, 1:50) for 30 min at 37 °C in a humid chamber. Slides were then washed three times for 5 min in PBS-Triton, incubated with an anti-ssDNA (Millipore, 1:50) for 30 min at 37 °C in a humid chamber, washed three times in PBS-Triton and incubated with secondary antibody anti-mouse IgG2A 647 (Thermo Fisher Scientific, 1:50) for 30 min at 37 °C in a humid chamber. After three final washes in PBS-Triton, slides were air-dried and mounted with 20 μl of Prolong Gold antifade reagent (Invitrogen life technology). Images were then acquired on Celldiscoverer 7 at 50X and analyzed with Zen3.0 to measure CldU tracks when preceded to an IdU track in all conditions.

### Protein purification

Flag (DYKDDDDK)-His Tag-Exo1 or (DYKDDDDK)-His Tag-Exo1(D173A) were generated by Q5 Site-Directed Mutagenesis Kit (New England Biolabs) with primers (forward primer: 5′-GAGGACT CGGCTCTCCTAGCT-3′, reverse primer: 5′-TGTAATTATGGCTTGC ACAATTC-3′) and cloned in the pFASTBAC plasmid (Invitrogen). Flag-His Tag-EXO1 and Flag-His Tag-EXO1(D173A) were expressed in Sf9 insect cells through recombinant baculovirus using Bac to

Bac Expression System (Invitrogen). Seventy-two hours after infection, Sf9 cells were harvested, and pellets were lysed in lysis buffer (50 mM Tris-HCl pH7.5, 150 mM NaCl, 1 mM EDTA, 10% Glycerol, 1 mM DTT, 0.05% TritonX-100 with protease inhibitors) and homogenized by gentle pipetting. The cell lysate was sonicated on ice for 30 s for five times and incubated with 1 mM $MgCl_2$ and 15 U/mL benzonase nuclease for 1 h at 4 °C followed by centrifugation at 35000 rpm for 1 h at 4 °C. The soluble cell lysate was incubated with 600 μl Anti-FLAG M2 Affinity Gel (Sigma-Aldrich) for 2 h at 4 °C with rotation. The beads were washed twice with Flag-binding buffer (50 mM Tris-HCl pH7.5, 150 mM NaCl, 1 mM EDTA, 10% Glycerol, 1 mM DTT, 0.025% Triton X-100) followed by incubation with 10 ml HSP buffer (Flag-binding buffer with 5 mM ATP, 15 mM $MgCl_2$) for 45 min at 4 °C with rotation. After three washes in Flag-washing buffer (50 mM Tris-HCl pH7.5, 250 mM NaCl, 1 mM EDTA, 10% Glycerol, 1 mM DTT, 0.025% Triton X-100) and one wash with Flag-elution buffer (50 mM Tris-HCl pH7.5, 150 mM NaCl, 10% Glycerol, 1 mM DTT, 0.025% Triton X-100), the protein was eluted twice with 500 μg/mL 3XFLAG peptide for 45 min at 4 °C. The elution was incubated with TALON® Metal Affinity Resin (Clontech) in P5 buffer (20 mM $Na_2HPO_4$ pH 7.0, 20 mM $NaH_2PO_4$ pH 7.0, 500 mM NaCl, 10% glycerol, 0.05% Triton-X-100 and 5 mM imidazole) for 1 h at 4 °C. The beads were washed once in P30 (5% P500 buffer complemented with 95% P5 buffer) followed by another wash with P80 buffer (85% P500 buffer complemented with 15% P5 buffer). The protein was eluted by incubating with P500 buffer (20 mM $Na_2HPO_4$ pH 7.0, 20 mM $NaH_2PO_4$ pH 7.0, 500 mM NaCl, 10% glycerol, 0.05% Triton-X-100 and 500 mM imidazole) for 2 min at 4 °C and dialyzed in EXO1 storage buffer (20 mM Tris-Acetate pH8, 200 mM $CH_3COOK$, 10% glycerol, 1 mM DTT). The purified Flag-His Tag-EXO1 and Flag-His Tag-EXO1(D173A) were stored in aliquots at −80 °C.

### In vitro DPC assays

Assays were performed by using probes generated from the plasmids (received from Dr. Julien Duxin, University of Copenhagen) where the DNA methyltransferase HpaII (M.HpaII, ~45 kDa) is or is not covalently linked to its recognition site, CCGG, via 5-fluoro-2′-deoxycytosine on the top strand of DNA strands[67]. The plasmids were linearized by BssHII and labeled 3′ end with [α-$^{32}$P]-CTP using Klenow enzyme, generating a DPC probe that contains a DPC ~200-219 bp far from the 5′ DNA end as well as a control (CTL) probe without DPC. In vitro reactions were conducted using the CTL or DPC probe in standard buffer (20 mM HEPES pH7.5, 0.1 mM DTT, 0.05% TritonX-100, 100 μg/mL BSA) with 2 mM ATP and 5 mM $MgCl_2$. For the reactions with only EXO1, they were initiated on ice by adding 6 nM purified EXO1 or EXO1(D173A) described as above and transferred immediately to 37 °C for 2 h. For the reactions with the EXO1 whole resection machinery, the order of addition and incubation of the respective protein components were: MRN (10 nM, 5 min), RPA (100 nM, 5 min), BLM (15 nM, 3 min) and EXO1 (6 nM, 2 h) at 37 °C[46]. The reactions were stopped by proteinase K (Thermo Fisher Scientific) treatment for 30 min at 37 °C. Products were analyzed on a 1% native agarose gel. Gels were dried on DE81 paper (Whatman) and signals were detected by autoradiography. Densitometric analyses were performed using the FLA-5100 phosphorimager (Fujifilm) and quantified using the Image Reader FLA-5000 V1.0 software.

### Cell cycle analysis

150,000 RPE-1 cells were seeded into one well of a six-well plate for 18 h and treated with 25 ng/ml Mitomycin C (Sigma) or culture medium for 24 h. Cells were then harvested, washed once with PBS, and fixed in 70% EtOH. After two washes in 5 ml 1% BSA-PBS with a centrifugation at 1000 rpm 4 °C for 5 min, the cells were stained in

PBS with 50 μg/ml propidium iodide (Sigma-Aldrich) and 0.1 μg/μl RNaseA (New England Biology) for 30 min at room temperature. At least 15,000 cells for each condition were analyzed in the BD Accuri™ C6 Plus flow cytometer using the BD Accuri™ C6 plus software.

## Metaphase spread

MEFs (WT and KO) were seeded into 10 cm dish with a density of 150,000 cells per dish. Mitomycin C (Sigma) treatment was performed for 48 h. Before collection, cells were treated with 1 μM of nocodazole (Sigma) for 3 h at 37 °C. The medium was then removed slowly and kept in 15 ml centrifuge tubes, cells were washed with 2 ml PBS (Wisent Bio Products) and kept in the same 15 ml centrifuge tubes. One milliliter of trypsin was applied to the cells for maximum 1 min, and cells were centrifuged at 1500 rpm for 5 min. The pellets were resuspended in 0.5 ml hypotonic solution (14% FBS, 10.7 mM KCl in $H_2O$), followed by the addition of 6.5 ml hypotonic solution. The tubes were placed at 37 °C for 15 min, and fixed with 10 drops of fixative solution (EtOH:acetic acid 3:1). Cells were centrifuged at 1500 rpm for 5 min at 4 °C. Most of the supernatant was removed and the pellets were resuspended by tapping. Seven milliliters of fixative solution was added to each tube and the tubes were kept at 4 °C for 48 h. The tubes were centrifuged at 1500 rpm for 5 min at 4 °C. The pellets were resuspended in 100–500 μl fixative solution (depending on their size), following by quickly spreading on prechilled slides. Coverslips were mounted onto slides with ProLong® Gold Antifade Mountant with DAPI (Invitrogen life technology).

## DPC detection

DPCs in RPE-1 WT and EXO1 KO (KO 7, KO 11) cells with indicated treatment were isolated using a modified rapid approach to DNA adduct recovery (RADAR) assay[68,69]. Total DPCs were visualized by SilverQuest™ Silver Staining Kit (Invitrogen) as recommended by the manufacturer after electrophoretic separation on 4–12% Bis-Tris polyacrylamide gels.

## Reporting summary

Further information on research design is available in the Nature Portfolio Reporting Summary linked to this article.

## Data availability

Source data are provided as a Source Data file. All materials in the manuscript can be obtained upon request. Source data are provided with this paper.

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

## Acknowledgements

The authors thank the Masson lab for critically reviewing the manuscript and for helpful comments. We thank Agata Smogorzewska, Detlev Shindler, Annamaria Ruggiano and Kristijan Ramadan, as well as Fatemeh Mashayekhi and Ismail Ismail for technical advice, Winfried Edelmann for Exo1 deficient MEFs, Josephine Dorsman for VU1365 and RPE2 cells, Alan d'Andrea for cell lines, Tom Moss for the use of a Leica SP5 microscope, and Yannick Doyon for the AAVS1 complementation system. L.G.S. was

a FRQS postdoctoral fellow, and Y.G. was supported by Fondation du CHU de Québec and FRQS Ph.D. scholarships. This work was funded by grants from the European Research Council (grant no. 715975-DPC_REPAIR) to J.P.D and N.B.L., CIHR MOP-152948 to A.F.T., NSERC grant number RGPIN-2016-05847 to S.H. and CIHR FDN-388879 to J.Y.M. S.H. is an FRQS Junior 2 scholar, and A.F.T. is a Canada Research Chair in Molecular Virology and Genomic Instability and is supported by the Foundation J.-Louis Lévesque. J.Y.M. is a Tier I Canada Research Chair in DNA repair and Cancer Therapeutics.

## Author contributions

The two co-first authors, Y.G. and L.G. contributed to project conception, performed and analyzed experiments, generated related material and co-wrote the paper. J.D. performed the bioinformatic analysis of the CRISPR-Cas9 screen; A.B. worked in parallel with Y.C. to amplify the TKO_v1 library and generate lentiviruses. A.R. and L.M. performed experiments. N.B.L. (under the supervision of J.P.D.) generated the M.HpaII linked DPC substrates. A.F.T. supervised A.B. and co-supervised J.D. with S.H. J.Y.M. contributed to project conception, coordination, supervision, and co-wrote the paper.

## Competing interests

The authors declare no competing interests.
