## [Peer Review File · Nature Communications]

A CRISPR-Cas9 screen identifies EXO1 as a formaldehyde resistance geneREVIEWER COMMENTS

Reviewer #1 (Remarks to the Author):

Summary: Using a CRISPR-Cas9 screen, the authors identified EXO1 as an important genetic determinant for formaldehyde resistance. They went on to characterize a number of cell-based phenotypes, including parameters of the replication stress response, and explored the potential role of EXO1 in repair of ICLs or DPCs induced by formaldehyde exposure. Given that EXO1 is a prominent player in DNA repair and genome maintenance, the studies will attract some attention from the field. However, it was previously shown that EXO1 deficiency sensitizes cells to formaldehyde (see below); consequently, that aspect is not novel. As mentioned below, there are concerns that more mechanistic studies might have elucidated a better understanding of EXO1's proposed role in replication stress caused by formaldehyde exposure. There are other experimental concerns raised, and the manuscript overall requires considerable proofreading and evaluation of the accuracy of statements as written.

Specific Critical Comments:

There are numerous grammatical errors in the manuscript. For example, the last sentence of the abstract says: "Collectively, we show that EXO1 limits replication stress and DNA damage to counteracts formaldehyde-induced genome instability." "to counteracts" should be "to counteract". The manuscript needs to be carefully proofread for errors in writing, which appear throughout the text.

Odd phrasing and difficult to read:

"Further advances on how aldehydes may affect FA were when inactivation of ALDH2, which oxidizes acetaldehyde to acetic acid, was shown to exacerbate FA phenotypes 20 21."

Unclear why terms are capitalized:

It was also reported that aged *Aldh2*^{-/-} *Fancd2*^{-/-} mice that do not develop leukemia, suffer aplastic anemia with the accumulation of DNA damage in the Hematopoietic Stem and Progenitor Cell pool 22.

Are there disadvantages to using hTERT-immortalized RPE-1 cells (RPE-1) for the CRISPR-Cas9 screen? How relevant is this cell line to FA? Was the screen validated in at least one other cell line to assess the universality of the identified hits?

It is critical that the authors make a compelling and cohesive argument for the significant advance in their findings as compared to those previously showing that EXO1 depletion resulted in formaldehyde sensitivity. See refs below:

26. Zhao, Y. et al. Applying genome-wide CRISPR to identify known and novel genes and pathways that modulate formaldehyde toxicity. *Chemosphere* 269, 128701 (2021).

27. Olivieri, M. et al. A Genetic Map of the Response to DNA Damage in Human Cells. *Cell* 182, 481-496 e421 (2020).

DNA replication stress induced by formaldehyde should also be examined by changes to RPA phosphorylation, as a major component of the checkpoint response and signaling.

The decrease in CldU track length under formaldehyde treatment in EXO1-depleted cells was interpreted to suggest a role of EXO1 at DNA replication forks. However, this could have been examined more extensively by an assessment of nascent strand degradation, fork asymmetry, dormant origin firing, etc. Surprisingly, for a study in which the authors conclude in the Abstract that "EXO1 limits replication stress and DNA damage to counteract formaldehyde-induced DNA damage", the replication fork dynamics as a function of EXO1 status were not characterized in a detailed or mechanistic manner. For example, one could have readily assessed using the DNA fiber technique if the apparent replication problems in EXO1-deficient cells could have been enhanced or suppressed by establishing cellular co-deficiency of DNA translocases or structure-specific nucleases or implicated in regressed fork formation and processing.

The elevated gammaH2AX staining of EXO-1 depleted cells exposed to formaldehyde would

suggest the accumulation of DSBs. It would have been informative to assess if these DSBs were dependent on the structure-specific nuclease Mus81, which acts at regressed forks and can lead to chromosomal instability. However, the authors did not do this fairly routine experiment, leaving the mechanistic basis for the DSB formation in question.

Were the gammaH2AX foci enriched or specifically found in replicating cells? Co-staining with PCNA or some other markers for S-phase cells might have been informative and support some of the authors' suggestions.

A series of experiments were performed to investigate if EXO1 acts on ICLs and/or DPCs. This is a reasonable line of inquiry because cellular exposure to formaldehyde can induce both types of lesions. What seems odd (unless I missed it) is that the authors did not determine by genetic studies if EXO-1 and various FA genes are epistatic or not for formaldehyde sensitivity and more specifically repair of ICLs and/or DPCs. So as to not elude about this comment, I would have liked to have seen if the co-deficiency of EXO1 and a classic FA gene in the core complex, or FANCD2/I, or a downstream player in the FA ICL repair pathway showed similar or different sensitivity to formaldehyde as compared to the singular deficiency.

Sometimes I find slightly oversimplistic or not entirely the best or most accurate phrasing at various places in the manuscript. For example, it is written in the Discussion: "EXO1 is a 5' to 3' double-stranded DNA exonuclease 29" ; here ref. 29 is a review. This might also be interpreted to mean that EXO1 only acts on blunt-ended dsDNA; while EXO1 does indeed act upon blunt duplex DNA Molecular interactions of human Exo1 with DNA - PubMed (nih.gov), it also acts on other substrates. For example, EXO1 possesses an intrinsic structure-specific flap endonuclease type activity. Human exonuclease 1 (EXO1) activity characterization and its function on flap structures (nih.gov)

Clarification in what is written will avoid misunderstandings from the readership. This issue is not unique to this particular case. I found questionable or slightly oversimplistic or outright confusing statements in other places in the manuscript.

Figures:

Please show the DNA substrate (and location of radiolabel) used for biochemical studies shown in Fig. 6e.

It is odd in Fig. 7 and Fig. 8 that formaldehyde concentrations are expressed in molar terms whereas MMC is expressed in ng/ml.

Remove the term "concentration" in x-axis labels for Figs. 6a, 8a, 8b

Reviewer #2 (Remarks to the Author):

The manuscript is generally well presented and the results support the main conclusions.

The authors performed a screen to identify new proteins involved in cellular stress responses activated by formaldehyde. They have identified a EXO1 has a DNA repair protein involved in ICL and DPC repairs and presented results suggesting the involvement of EXO1 in the early steps of the FA pathway.

Minor comments

Page 6 "While the latter observation was surprising". Is it really surprising? In my understanding of IR, the DSB monitored is the direct consequence of the irradiation and not the consequences of the collapse of a RF like for formaldehyde.

Page 7 PUVA is the current standard method to monitor the recruitment of proteins to ICLs. TMP induces mainly ICLs if the right conditions of UVA are used. Did the authors use Angelicin as a control? Is the recruitment equivalent in all cells? The fact that a recruitment is observed by PUVA may suggest a recruitment that is not replication dependent. The recruitment can be mediated by MUTS α as described in Kato et al. 2017. Can authors integrate this point in the discussion?

Page 7-8 The authors have convincingly showed the activity of EXO1 to process in vitro. While it is not necessary to have similar approach for the processing of ICLs since it is known, it would have been appreciated to have the papers cited.

Page 8 " We can conclude that EXO1 process DBCs, including TOP1cc" should be replace by "We can conclude that EXO1 process DBCs in vitro, including TOP1cc

Page 8 The differences in FancD2 foci is strong but is not sufficient to show a defect in the FA pathway activation. The ubiquitination of FANCD2 should be monitored by WB.

Page 8. A 4N increased upon exposure to ICLs is an hallmark of FA and should be analyzed in EXO1 deficient cells

Discussion:

Can authors discuss:

-The potential redundancy with other ICL repair nucleases (FAN1?). Since it has been shown that EXO1 have redundancy with many nuclease regarding it function in MMR it may be the case for ICL.

-The use of ADLH2/EXO1 double KO mice to investigate whether EXO1 can be a FA protein?

The following references should be integrated in the discussion

Raschle et al 2015 for the recruitment of EXO1 to psoralen

Whao et al. 2021 for the use of CRISPR screen involving formaldehyde

Major comment

Genetic epistasis should be done to determine if FA EXO1 double KO are more sensitive to Formaldehyde than single mutant. This will confirm that EXO1 act in the FA pathway.

Reviewer #3 (Remarks to the Author):

The authors used CRISPR screening to identify cellular modulators of formaldehyde sensitivity. They used RPE1-p53 cells expressing CAS9 and carried out screens that identified a small network of proteins that increased sensitivity when targeted. This included ADH5 and ESD, as well as numerous repair proteins including EXO1, that was previously seen to be a sensitizer in screens carried out in the same cell line (Olivieri et al). They validated this hit using KO cell lines and complementation. Notably, this was not mimicked by depletion of DNA2 that acts redundantly with EXO1 in resection of DSBs. They show that cells lacking EXO1 show increased CHK1 activation and slower fork speed upon formaldehyde treatment. This is followed by a titration analysis showing the formaldehyde induces replication stress, consistent with the previous data. This is extended to the EXO1 KO cells, where they see that RS is increased, with more DSBs and ssDNA accumulation, but there is little effect on the response to IR. They then show that EXO1 is recruited to formaldehyde damage and phosphorylated on an ATR site. To determine if EXO1 acted on ICLs or DPCs, they examined MMC sensitivity, that was modestly increased, and recruitment to UV laser damage that was independent of the exonuclease activity. To examine DPCs, they used staining for TOP1ccs and found an increase in the formaldehyde treated KO cells and they found that EXO1 activity was important for in vitro resection of DPC modified breaks. As this suggested EXO1 plays a role in both ICL and DPC repair, they examined FA pathway induction using FANCD2 as a marker and found that its recruitment was diminished in EXO1 KOs treated with MMC and that EXO1 recruitment was not impacted by FANCA deletion. Finally, they examined EXO1 deficient MEFs and see that they are sensitive to MMC and to Formaldehyde, consistent with previous results. In addition, metaphase analysis of MMC treated cells showed that EXO1 KO led to higher numbers of breaks and quadraradials, a characteristic of defective FA activity.

The manuscript is well organized and easy to follow and the data is generally of high quality. The novelty of some results is impacted by previous publications but several experiments do clarify

some issues raised in previous work, including non catalytic functions and DNA2 redundancy. Overall, I think the work will be of general interest to the DNA repair community and provides clear support and mechanistic detail for the role of EXO1 in formaldehyde repair. I think there are some strong conclusions that remain to be fully substantiated and have a few comments and suggestions.

1. Fork progression is assayed and no examples of raw data are provided anywhere (Figure 3).
2. The data indicating a role for EXO1 in TOP1cc (DPC) repair is a bit confusing with regards to previous data. Screens in the same cell line (Olivieri et al) reported that EXO1 loss makes cells more resistant to CPT treatment, in accordance with another report in MEFs (Rein et al, NAR 2015) and in contrast to a different paper using MEFs (Schaetzlein et al, 2013). I realize that the assay reported here is using formaldehyde, not CPT, but I am unclear how TOP1ccs would be fundamentally different in that they would require EXO1 in one case and its activity may be detrimental in another. Have the authors looked at CPT sensitivity, DPC accumulation (ex RADAR assay) or the DDR in this context to understand what the potential difference would be? I remain fairly unconvinced that EXO1 plays a key role in DPC removal from the data presented.
3. The authors generally do a good job of reporting for reproducibility but this is sometimes incomplete. For example the TOP1cc assay is reported as n=3 and individual cell data is shown but it is unclear if all data is combined or this is 1 representative experiment. In our experience, due to signal intensity variation between experiments, this is not usually straightforward to mix together and requires some normalization for statistics. So what was done here, or in any similar experiments, should be described in full detail for the reader to evaluate. I am not sure the statistics were performed correctly here, but lack necessary information to assess it.
4. No statistical analyses have been applied to survival curves (ex 2d, 6a). As some of these look pretty borderline, it would be informative to have this information. I am not suggesting that data need a p-value to be reported or interpreted but it provides additional confidence to some results, like MMC sensitivity, that are not very robust.
5. The authors make a strong point that EXO1 is upstream of the FA pathway. It would therefore be useful to see how EXO1 compares in the same experiment to an FA pathway KO (ex FANCA or FANCD2). While the data presented seems robust, I am also not entirely convinced that EXO1 controls the FA pathway as dramatically as proposed. Additional endpoint assays to look at relative survival in single and double KOs, to ensure epistasis, would help to substantiate some of these claims. It is notable that the sensitivity profiles of EXO1 and FA in screening do not overlap outside of formaldehyde and in response to formaldehyde, FA KOs scored much higher than EXO1 (Olivieri et al)- for example FANCI, L, E, D2, A etc. I therefore feel that in order to make the claim that EXO1 is apical, the authors need some direct comparisons and epistasis analysis.

Rebuttal NCOMMS-22-03197
A CRISPR-Cas9 screen identifies EXO1 as a formaldehyde resistance gene

Reviewer #1 (Remarks to the Author):

Summary: Using a CRISPR-Cas9 screen, the authors identified EXO1 as an important genetic determinant for formaldehyde resistance. They went on to characterize a number of cell-based phenotypes, including parameters of the replication stress response, and explored the potential role of EXO1 in repair of ICLs or DPCs induced by formaldehyde exposure. Given that EXO1 is a prominent player in DNA repair and genome maintenance, the studies will attract some attention from the field. However, it was previously shown that EXO1 deficiency sensitizes cells to formaldehyde (see below); consequently, that aspect is not novel. As mentioned below, there are concerns that more mechanistic studies might have elucidated a better understanding of EXO1's proposed role in replication stress caused by formaldehyde exposure. There are other experimental concerns raised, and the manuscript overall requires considerable proofreading and evaluation of the accuracy of statements as written.

Specific Critical Comments:

There are numerous grammatical errors in the manuscript. For example, the last sentence of the abstract says: "Collectively, we show that EXO1 limits replication stress and DNA damage to counteracts formaldehyde-induced genome instability." "to counteracts" should be "to counteract". The manuscript needs to be carefully proofread for errors in writing, which appear throughout the text.

Odd phrasing and difficult to read:

"Further advances on how aldehydes may affect FA were when inactivation of ALDH2, which oxidizes acetaldehyde to acetic acid, was shown to exacerbate FA phenotypes 20 21."

Unclear why terms are capitalized:

It was also reported that aged Aldh2^{-/-} Fancd2^{-/-} mice that do not develop leukemia, suffer aplastic anemia with the accumulation of DNA damage in the Hematopoietic Stem and Progenitor Cell pool 22.

REPLY: Thank you for the comments on our manuscript. We corrected the text accordingly. Careful manuscript proofreading was performed by a native English speaker and scientific editor.

Are there disadvantages to using hTERT-immortalized RPE-1 cells (RPE-1) for the CRISPR-Cas9 screen? How relevant is this cell line to FA? Was the screen validated in at least one other cell line to assess the universality of the identified hits?

REPLY: hTERT-immortalized RPE-1 cells (RPE-1) are a retinal pigment epithelium cell line. hTERT-RPE1 cells are genetically stable diploid cells widely used to model cell division and DNA repair in CRISPR-Cas9 screens. These cells were obtained from Dan Durocher in Toronto who published several screens with this cell line including Cell 182, 481-496 e421¹. It is relevant to Fanconi anemia as they can present microcornea, microphthalmia, ptosis, steep corneal curvatures,

small optic discs, ptosis and delay in visual processing skills ². We also confirmed that EXO1 leads to sensitivity to formaldehyde in Mouse Embryonic Fibroblasts (Figure 10) and performed an epistatic analysis in the squamous cell line VU1365 derived from a FA patient (new Suppl. Figure 9b).

It is critical that the authors make a compelling and cohesive argument for the significant advance in their findings as compared to those previously showing that EXO1 depletion resulted in formaldehyde sensitivity. See refs below:

26. Zhao, Y. et al. Applying genome-wide CRISPR to identify known and novel genes and pathways that modulate formaldehyde toxicity. *Chemosphere* 269, 128701 (2021).

27. Olivieri, M. et al. A Genetic Map of the Response to DNA Damage in Human Cells. *Cell* 182, 481-496 e421 (2020).

REPLY: In the initial submission we did cite the two papers ^{1 3}. In addition, we showed the shared hits from our CRISPR-Cas9 screen and the screen by Olivieri et al, presented by STRING (Suppl. Fig 1C). CRISPR screens generate long lists of targets in dire need of detailed functional assessment. From the previous references there was no follow-up on EXO1 and mechanistic studies, so we think that our current manuscript represents a significant advance in the FA field. As mentioned by the reviewer: "Given that EXO1 is a prominent player in DNA repair and genome maintenance, the studies will attract some attention from the field."

DNA replication stress induced by formaldehyde should also be examined by changes to RPA phosphorylation, as a major component of the checkpoint response and signaling.

REPLY: As suggested by the reviewer, we have monitored RPA S4/8 phosphorylation (as a marker for the induction of the DNA damage response and DNA resection) in EXO1 KO cells following formaldehyde treatment (new Fig. 6b). Unlike IR-treated cells, RPA S4/8 phosphorylation is increased in EXO1 KO cells following formaldehyde treatment. This is also in line with the increase of RPA foci (Fig. 6A) and increased strand degradation in DNA replication fork experiments after formaldehyde treatment (Figure 3c).

The decrease in CldU track length under formaldehyde treatment in EXO1-depleted cells was interpreted to suggest a role of EXO1 at DNA replication forks. However, this could have been examined more extensively by an assessment of nascent strand degradation, fork asymmetry, dormant origin firing, etc. Surprisingly, for a study in which the authors conclude in the Abstract that "EXO1 limits replication stress and DNA damage to counteract formaldehyde-induced DNA damage", the replication fork dynamics as a function of EXO1 status were not characterized in a detailed or mechanistic manner. For example, one could have readily assessed using the DNA fiber technique if the apparent replication problems in EXO1-deficient cells could have been enhanced or suppressed by establishing cellular co-deficiency of DNA translocases or structure-specific nucleases or implicated in regressed fork formation and processing.

REPLY: EXO1/MRE11 interplay at forks is well-established ⁴. To better detail replication fork dynamics as a function of EXO1 status we performed DNA fiber assays to assess nascent strand degradation. For nascent strand degradation, we used BRCA2 knockdown as a control which led to

nascent strand degradation in the presence of HU as published previously (new Suppl. Fig. 3a). We incorporated both IdU and CldU first and then added formaldehyde for 4 hours to observe a decrease in CldU tracks length decrease. First, as a control, we show that the distribution of IdU track length was similar between wild-type and EXO1 KO cells (new Suppl. Fig. 3e). In untreated cells, EXO1 inactivation led to a slight increase in strand degradation suggesting a protective role for EXO1 in normal conditions which was enhanced further by formaldehyde treatment (new Fig. 3c). Hence, in two different replication assays (assessment of CldU length after formaldehyde treatment and nascent strand degradation) EXO1 protects from replication stress and degradation caused by formaldehyde. As for the comment on cellular co-deficiency, we knockdown Mus81 (see next comment) and saw that knockdown of Mus81 is required to avoid the accumulation of DSBs and fork collapse after replication perturbation by formaldehyde when EXO1 is depleted.

The elevated gammaH2AX staining of EXO-1 depleted cells exposed to formaldehyde would suggest the accumulation of DSBs. It would have been informative to assess if these DSBs were dependent on the structure-specific nuclease Mus81, which acts at regressed forks and can lead to chromosomal instability. However, the authors did not do this fairly routine experiment, leaving the mechanistic basis for the DSB formation in question.

REPLY: This is very important indeed, as it was previously shown that Mus81 promotes the conversion of ICLs to DSBs in S-phase⁵. We checked whether the elevated gammaH2AX staining of EXO1 depleted cells exposed to formaldehyde depends on Mus81 (new Figure 5c-d). Indeed, the knockdown of Mus81 in EXO1 depleted cells led to a decrease in the number of γ -H2AX foci. Thus, knockdown of Mus81 is required to avoid the accumulation of DSBs and fork collapse after replication perturbation by formaldehyde when EXO1 is depleted.

Were the gammaH2AX foci enriched or specifically found in replicating cells? Co-staining with PCNA or some other markers for S-phase cells might have been informative and support some of the authors' suggestions.

REPLY: The γ -H2AX foci were enriched and specifically found in replicating cells as we did include EdU staining as a marker for S-phase cells (Figure 5, Suppl. Figures 4-5-6).

A series of experiments were performed to investigate if EXO1 acts on ICLs and/or DPCs. This is a reasonable line of inquiry because cellular exposure to formaldehyde can induce both types of lesions. What seems odd (unless I missed it) is that the authors did not determine by genetic studies if EXO-1 and various FA genes are epistatic or not for formaldehyde sensitivity and more specifically repair of ICLs and/or DPCs. So as to not elusive about this comment, I would have liked to have seen if the co-deficiency of EXO1 and a classic FA gene in the core complex, or FAND2/I, or a downstream player in the FA ICL repair pathway showed similar or different sensitivity to formaldehyde as compared to the singular deficiency.

REPLY: This was also a point raised by reviewers 2 and 3. We performed epistatic analysis in RPE2 cells KO for FANCA. Knockdown of EXO1 in FANCA^{-/-} cells led to additive formaldehyde sensitivity compared to FANCA^{-/-} alone. Thus, this suggests that EXO1 does not function epistatically with FANCA. Hence, we suggest that they work in parallel (but also overlapping also as EXO1 functions

in HR downstream of the FA pathway) pathways leading to the repair of formaldehyde-induced lesions (new Figure 8b in RPE2 cells). The same result was observed in FANCA-deficient VU1365 cells (new Suppl. Figure 9b). Thus, this experiment helped to refine our model (see new Figure 10e) where the burden of DPCs/ICLs are taken upon EXO1 or the FA pathway, but both cannot compensate for each other.

Sometimes I find slightly oversimplistic or not entirely the best or most accurate phrasing at various places in the manuscript. For example, it is written in the Discussion: "EXO1 is a 5' to 3' double-stranded DNA exonuclease 29" ; here ref. 29 is a review. This might also be interpreted to mean that EXO1 only acts on blunt-ended dsDNA; while EXO1 does indeed act upon blunt duplex DNA Molecular interactions of human Exo1 with DNA - PubMed (nih.gov), it also acts on other substrates. For example, EXO1 possesses an intrinsic structure-specific flap endonuclease type activity. Human exonuclease 1 (EXO1) activity characterization and its function on flap structures (nih.gov)

REPLY: Point taken. Reference 29 was removed. It now reads as follows: "EXO1 is a 5' to 3' double-stranded DNA exonuclease with modest endonuclease or 5' flap activity ⁶. Its strong capacity for degrading double-stranded DNA must be restrained in some conditions".

Clarification in what is written will avoid misunderstandings from the readership. This issue is not unique to this particular case. I found questionable or slightly oversimplistic or outright confusing statements in other places in the manuscript.

REPLY: The manuscript has been proofread by a native English speaker and scientific editor.

Figures:

Please show the DNA substrate (and location of radiolabel) used for biochemical studies shown in Fig. 6e.

REPLY: This is found in Suppl. Figure 8a-b and the associated figure legends. The green asterisk denotes the position of the radiolabel on the substrate.

It is odd in Fig. 7 and Fig. 8 that formaldehyde concentrations are expressed in molar terms whereas MMC is expressed in ng/ml.

Remove the term "concentration" in x-axis labels for Figs. 6a, 8a, 8b

REPLY: MMC is now expressed in molar terms and we removed the term concentration.

Reviewer #2 (Remarks to the Author):

The manuscript is generally well presented and the results support the main conclusions. The authors performed a screen to identify new proteins involved in cellular stress responses activated by formaldehyde. They have identified a EXO1 has a DNA repair protein involved in ICL and DPC repairs and presented results suggesting the involvement of EXO1 in the early steps of the FA pathway.

REPLY: Thank you for the positive assessment of our manuscript and suggestions for the discussion.

Minor comments

Page 6 “While the latter observation was surprising”. Is it really surprising? In my understanding of IR, the DSB monitored is the direct consequence of the irradiation and not the consequences of the collapse of a RF like for formaldehyde.

REPLY: The reviewer is right. However, we were expecting that some of the DSBs from IR result in replication fork collapse and hence some sensitivity to IR. It is now phrased as follows: “The latter observation is consistent with the absence of significant IR sensitivity in EXO1 KO clones compared to WT cells (Suppl. Figure 5a). This might be explained by the fact that DNA2 can compensate for the absence of EXO1 ⁷”

Page 7 PUVA is the current standard method to monitor the recruitment of proteins to ICLs. TMP induces mainly ICLs if the right conditions of UVA are used. Did the authors use Angelicin as a control? Is the recruitment equivalent in all cells? The fact that a recruitment is observed by PUVA may suggest a recruitment that is not replication dependent. The recruitment can be mediated by MUTSa as described in Kato et al. 2017. Can authors integrate this point in the discussion?

REPLY: Upon TMP pretreatment followed by UV irradiation, the near totality of cells expressing GFP-EXO1 WT or the GFP-EXO1 D173A mutant displayed recruitment, with varying intensities between individual cells. We did observe some recruitment outside of S-phase, consistent with the replication independent ICL repair pathway. We have incorporated this in the discussion: “We noticed that the recruitment of EXO1 TMP-induced ICLs also occurred outside of S-phase. This is consistent with a MMR-dependent ICL repair mechanism which requires the nuclease activities of MutL α and EXO1 ⁸.”

As suggested, we monitored the recruitment of EXO1 following treatment with angelicin, a photoreactive psoralen analogue, which unlike TMP can form only monoadducts and not ICLs upon UV irradiation. The irradiation conditions were set using CSB as a control, a protein known to accumulate at sites of angelicin monoadducts (?). Under conditions where CSB is recruited in the presence of angelicin and not with the laser alone, none of the cells showed EXO1 recruitment, suggesting EXO1 responds to ICL rather than monoadducts. Data were added in new Suppl. Figure 7b.

Page 7-8 The authors have convincingly showed the activity of EXO1 to process in vitro. While it is not necessary to have similar approach for the processing of ICLs since it is known, it would have been appreciated to have the papers cited.

REPLY: Indeed, the Gauthier group has shown that MutSa recruits MutLa and EXO1 to ICL lesions, and the catalytic activity of both these nucleases is essential for ICL repair⁸. This is also added in the discussion ⁸.

Page 8 " We can conclude that EXO1 process DBCs, including TOP1cc" should be replaced by "We can conclude that EXO1 process DBCs in vitro, including TOP1cc"

REPLY: Indeed. Done.

Page 8 The differences in FancD2 foci is strong but is not sufficient to show a defect in the FA pathway activation. The ubiquitination of FANCD2 should be monitored by WB.

REPLY: Thank you for this suggestion. Following experiments with epistatic analyses (a major point raised by all reviewers) we had to perform further analysis of FANCD2 activation in EXO1 deficient cells (new Figure 8a, new Suppl. Figure 9a). As depletion of FANCA and EXO1 lead to additive sensitivity to formaldehyde, this suggested that EXO1 does not function epistatically with FANCA. Hence, depletion of EXO1, leads to the accumulation of DPCs and DSBs in S-phase, which activates the FA pathway (FANCD2 formation new Figure 8a and a slight increase in FANCD2 L/S ratio by WB new Suppl. Figure 9a). This said, EXO1 also functions in HR downstream of the FA pathway, which may explain why the FA pathway cannot compensate for the absence of EXO1. Our model has been revised accordingly.

Page 8. A 4N increased upon exposure to ICLs is a hallmark of FA and should be analyzed in EXO1 deficient cells.

REPLY: As suggested by the reviewer, we performed this experiment and saw that EXO1 depletion in normal or formaldehyde conditions leads to an increase in 4N cells (new Figure 10d).

Discussion:

Can authors discuss:

-The potential redundancy with other ICL repair nucleases (FAN1?). Since it has been shown that EXO1 have redundancy with many nucleases regarding its function in MMR it may be the case for ICL.

REPLY: As suggested by the reviewer, we also investigated whether EXO1 shares some potential redundancy with other ICL repair nucleases such as FAN1 (new Suppl. Figure 9c). Co-depletion of FAN1 and FANCA, or FAN1 and EXO1 did not increase further formaldehyde sensitivity suggesting an epistatic relationship.

-The use of ADLH2/EXO1 double KO mice to investigate whether EXO1 can be a FA protein?

REPLY: This is an interesting idea but outside of the scope of this paper since it involves mouse models which can take years to characterize. However, we proposed this experiment in the discussion for future studies.

The following references should be integrated in the discussion

Raschle et al 2015 for the recruitment of EXO1 to psoralen

Whao et al. 2021 for the use of CRISPR screen involving formaldehyde

REPLY: We have cited both references in the manuscript.

Major comment

Genetic epistasis should be done to determine if FA EXO1 double KO are more sensitive to Formaldehyde than single mutant. This will confirm that EXO1 act in the FA pathway.

REPLY: This was also a point raised by reviewers 1 and 3. We performed epistatic analysis in RPE2 cells KO for FANCA. Knockdown of EXO1 in FANCA^{-/-} cells led to additive formaldehyde sensitivity compared to FANCA^{-/-} alone. Thus, this suggests that EXO1 do not function epistatically with FANCA. Hence, we suggest that they work in parallel (but also overlapping also as EXO1 functions in HR downstream of the FA pathway) pathways leading to the repair of formaldehyde-induced lesions (new Figure 8b in RPE2 cells). The same pattern was observed in FANCA-deficient VU1365 cells (new Suppl. Figure 9b). Thus, this experiment helped to refine our model (see new Figure 10e) where the burden of DPCs/ICLs are taken upon EXO1 or the FA pathway, but both cannot compensate for each other.

Reviewer #3 (Remarks to the Author):

The authors used CRISPR screening to identify cellular modulators of formaldehyde sensitivity. They used RPE1-p53 cells expressing CAS9 and carried out screens that identified a small network of proteins that increased sensitivity when targeted. This included ADH5 and ESD, as well as numerous repair proteins including EXO1, that was previously seen to be a sensitizer in screens carried out in the same cell line (Olivieri et al). They validated this hit using KO cell lines and complementation. Notably, this was not mimicked by depletion of DNA2 that acts redundantly with EXO1 in resection of DSBs. They show that cells lacking EXO1 show increased CHK1 activation and slower fork speed upon formaldehyde treatment. This is followed by a titration analysis showing the formaldehyde induces replication stress, consistent with the previous data. This is extended to the EXO1 KO cells, where they see that RS is increased, with more DSBs and ssDNA accumulation, but there is little effect on

the response to IR. They then show that EXO1 is recruited to formaldehyde damage and phosphorylated on an ATR site. To determine if EXO1 acted on ICLs or DPCs, they examined MMC sensitivity, that was modestly increased, and recruitment to UV laser damage that was independent of the exonuclease activity. To examine DPCs, they used staining for TOP1ccs and found an increase in the formaldehyde treated KO cells and they found that EXO1 activity was important for in vitro resection of DPC modified breaks. As this suggested EXO1 plays a role in both ICL and DPC repair, they examined FA pathway induction using FANCD2 as a marker and found that its recruitment was diminished in EXO1 KOs treated with MMC and that EXO1 recruitment was not impacted by FANCA deletion. Finally, they examined EXO1 deficient MEFs and see that they are sensitive to MMC and to Formaldehyde, consistent with previous results. In addition, metaphase analysis of MMC treated cells showed that EXO1 KO led to higher numbers of breaks and quadraradials, a characteristic of defective FA activity.

The manuscript is well organized and easy to follow and the data is generally of high quality. The novelty of some results is impacted by previous publications but several experiments do clarify

some issues raised in previous work, including non catalytic functions and DNA2 redundancy. Overall, I think the work will be of general interest to the DNA repair community and provides clear support and mechanistic detail for the role of EXO1 in formaldehyde repair. I think there are some strong conclusions that remain to be fully substantiated and have a few comments and suggestions.

REPLY: Thank you for your enthusiasm about our manuscript and we have toned down some conclusions. In particular, our model has been refined due to new experiments (epistatic analyses).

1. Fork progression is assayed and no examples of raw data are provided anywhere (Figure 3).

REPLY: We apologize for this omission. Distribution of fibers and examples are provided in new Suppl. Figure 3.

2. The data indicating a role for EXO1 in TOP1cc (DPC) repair is a bit confusing with regards to previous data. Screens in the same cell line (Olivieri et al) reported that EXO1 loss makes cells more resistant to CPT treatment, in accordance with another report in MEFs (Rein et al, NAR 2015) and in contrast to a different paper using MEFs (Schaezlein et al, 2013). I realize that the assay reported here is using formaldehyde, not CPT, but I am unclear how TOP1ccs would be fundamentally different in that they would require EXO1 in one case and its activity may be detrimental in another. Have the authors looked at CPT sensitivity, DPC accumulation (ex RADAR assay) or the DDR in this context to understand what the potential difference would be? I remain fairly unconvinced that EXO1 plays a key role in DPC removal from the data presented.

REPLY: The main point is indeed to assess DPC accumulation. We turned our attention to the Topcc assay which is the most reliable for us. We observed consistently that Topcc was increased (~2-fold) following EXO1 deletion (New Suppl. Fig. 8a). This was consistent with immunofluorescence data showing the same tendency (Fig. 7c).

As suggested by the reviewer, we have also used the RADAR assay to look at DPC accumulation in EXO1 KO cells. We have performed several assays which indicated that DPC accumulation (as measured by the smear density) is increased in EXO1 KO7 and EXO1 KO11 cells compared to WT cells following 100uM formaldehyde treatment (New Suppl. Fig 8b). However, there are some growing concerns in the field that the RADAR technique can be unreliable and subject to variations. Overall, EXO1 deletion leads to DPC accumulation using two methods (Topcc detection and RADAR assay). In the future, we look forward to using a more sensitive method for DPC measurement.

3. The authors generally do a good job of reporting for reproducibility but this is sometimes incomplete. For example the TOP1cc assay is reported as n=3 and individual cell data is shown but it is unclear if all data is combined or this is 1 representative experiment. In our experience, due to signal intensity variation between experiments, this is not usually straightforward to mix together and requires some normalization for statistics. So what was done here, or in any similar experiments, should be described in full detail for the reader to evaluate. I am not sure the statistics were performed correctly here, but lack necessary information to assess it.

REPLY: All data is the mean of 3 independent experiments and p-values are mentioned in Figure legends. We added statistical analysis for survival curves in Suppl. Table 2.

4. No statistical analyses have been applied to survival curves (ex 2d, 6a). As some of these look pretty borderline, it would be informative to have this information. I am not suggesting that data need a p-value to be reported or interpreted but it provides additional confidence to some results, like MMC sensitivity, that are not very robust.

REPLY: Statistical analyses are in figure legends and statistical analyses for survival curves are provided in Suppl. Table 2 to highlight key differences.

5. The authors make a strong point that EXO1 is upstream of the FA pathway. It would therefore be useful to see how EXO1 compares in the same experiment to an FA pathway KO (ex FANCA or FANCD2). While the data presented seems robust, I am also not entirely convinced that EXO1 controls the FA pathway as dramatically as proposed. Additional endpoint assays to look at relative survival in single and double KOs, to ensure epistasis, would help to substantiate some of these claims. It is notable that the sensitivity profiles of EXO1 and FA in screening do not overlap outside of formaldehyde and in response to formaldehyde, FA KOs scored much higher than EXO1 (Olivieri et al)- for example FANCI, L, E, D2, A etc. I therefore feel that in order to make the claim that EXO1 is apical, the authors need some direct comparisons and epistasis analysis.

REPLY: This was also a point raised by reviewers 1 and 2. We performed epistatic analysis in RPE2 cells KO for FANCA. Knockdown of EXO1 in FANCA-/- cells led to additive formaldehyde sensitivity compared to FANCA-/- alone. Thus, this suggests that EXO1 does not function epistatically with FANCA. Hence, we suggest that they work in parallel (but also overlapping also as EXO1 functions in HR downstream of the FA pathway) pathways leading to the repair of formaldehyde-induced lesions (new Figure 8b in RPE2 cells). The same pattern was observed in FANCA-deficient VU1365 cells (new Suppl. Figure 9b). Thus, this experiment helped to refine our model (see new Figure 10e) where the burden of DPCs/ICLs are taken upon EXO1 or the FA pathway, but both cannot compensate for each other.

REFERENCES

1. Olivieri, M. et al. A Genetic Map of the Response to DNA Damage in Human Cells. *Cell* **182**, 481-496 e421 (2020).
2. Tornquist, A.L., Martin, L., Winiarski, J. & Fahnehjelm, K.T. Ocular manifestations and visual functions in patients with Fanconi anaemia. *Acta Ophthalmol* **92**, 171-178 (2014).
3. Zhao, Y. et al. Applying genome-wide CRISPR to identify known and novel genes and pathways that modulate formaldehyde toxicity. *Chemosphere* **269**, 128701 (2021).
4. Lemacon, D. et al. MRE11 and EXO1 nucleases degrade reversed forks and elicit MUS81-dependent fork rescue in BRCA2-deficient cells. *Nat Commun* **8**, 860 (2017).
5. Hanada, K. et al. The structure-specific endonuclease Mus81-Eme1 promotes conversion of interstrand DNA crosslinks into double-strands breaks. *EMBO J* **25**, 4921-4932 (2006).
6. Keijzers, G., Bohr, V.A. & Rasmussen, L.J. Human exonuclease 1 (EXO1) activity characterization and its function on flap structures. *Biosci Rep* **35** (2015).

7. Karanja, K.K., Cox, S.W., Duxin, J.P., Stewart, S.A. & Campbell, J.L. DNA2 and EXO1 in replication-coupled, homology-directed repair and in the interplay between HDR and the FA/BRCA network. *Cell Cycle* **11**, 3983-3996 (2012).
8. Kato, N. et al. Sensing and Processing of DNA Interstrand Crosslinks by the Mismatch Repair Pathway. *Cell Rep* **21**, 1375-1385 (2017).
9. Iyama, T. & Wilson, D.M., 3rd Elements That Regulate the DNA Damage Response of Proteins Defective in Cockayne Syndrome. *J Mol Biol* **428**, 62-78 (2016).

REVIEWERS' COMMENTS

Reviewer #1 (Remarks to the Author):

The authors have systematically addressed many comments raised, and performed additional experiments that go a long way to improve the take-home messages and flesh out details of their findings.

Reviewer #2 (Remarks to the Author):

I thank the authors to have take into account all my questions.

The presented results are convincing and revised the models in frame with the presented data.

2 minors concerns:

The figure number is missing in the text when authors described epistasis data between FAN1 and Exo1.

A sentence to introduce the relevance of looking at the epistasis between the two genes should be present. i.e. these nucleases are redundant in author repair and it has been shown that FAN1 repair of ICL is mediated by its interaction with the MMR protein MLH1.

Finally I agree with authors about the fact that the requested experiment in mice is not relevant anymore since in the revised model EXO1 act in parallel to FA. Therefore I found confusing the discussion part on EXO1 and FA patient or mice experiment. I am not sure it had anything

Reviewer #3 (Remarks to the Author):

The authors have performed a number of new experiments that have helped to elucidate the role of EXO1 in FA repair and clarify its role in relation to the Fanconi pathway. These have improved the manuscript and provided some new insights. I have only a few minor comments.

1. Related to the new experiments that were performed following another reviewer's suggestion, I would like to point out that a protective effect of EXO1 loss on forks was reported previously (Rein et al, NAR 2015) after HU and CPT treatment. It was also reported that MUS81 co-depletion did not alter the sensitivity of EXO1 null cells to CPT. Given an opposite effect is observed with FA treatment, it would indicate there is some degree of specificity for the damage type that may be interesting to point out given the somewhat contradictory data addressing the role of EXO1 in damage sensitivity. The Rein paper also reports no IR sensitivity to EXO1 null cells, consistent with data shown in the paper and related to another reviewer comment regarding how "surprising" it is- it is published and could be referenced.

2. In our hands, the RADAR assay has been reliable in identifying TOP1 and TOP2ccs, but we and others are using a slot blotter to concentrate the signal. There are likely less DPCs with FA than CPT or Etoposide treatment. It would be useful to make sure that the data throughout the paper is labeled as TOP1cc and not TOPcc for clarity. In the rebuttal, the authors state "EXO1 deletion leads to DPC accumulation using two methods (Topcc detection and RADAR assay)" but the graph in Supp Figure 8b does not appear to show any difference in the RADAR assay. Please clarify this result- it was seen with 2 methods or it was seen with 1 method?

REVIEWERS' COMMENTS

Reviewer #1 (Remarks to the Author):

The authors have systematically addressed many comments raised, and performed additional experiments that go a long way to improve the take-home messages and flesh out details of their findings.

REPLY: We would like to thank the reviewer for his/her comments which improved the manuscript. We are happy with the final version of the paper.

Reviewer #2 (Remarks to the Author):

I thank the authors to have take into account all my questions.

The presented results are convincing and revised the models in frame with the presented data.

REPLY: Thank you. We would like to thank the reviewer for his/her comments which improved the manuscript.

2 minors concerns:

The figure number is missing in the text when authors described epistasis data between FAN1 and Exo1.

REPLY: We added the reference to Suppl. Fig. 9c.

A sentence to introduce the relevance of looking at the epistasis between the two genes should be present. i.e. these nucleases are redundant in author repair and it has been shown that FAN1 repair of ICL is mediated by its interaction with the MMR protein MLH1.

REPLY: We added the sentence: "We also investigated whether EXO1 shares some potential redundancy with other ICL repair nucleases, as FAN1 partially compensates for the lack of exonuclease 1 in mismatch repair ¹."

Finally I agree with authors about the fact that the requested experiment in mice is not relevant anymore since in the revised model EXO1 act in parallel to FA. Therefore I found confusing the discussion part on EXO1 and FA patient or mice experiment. I am not sure it had anything

REPLY: The discussion was slightly changed. "The generation of double knockout mice of Exo1 and a FA gene would be valuable to investigate the consequence of EXO1

deletion on the progression of the disease. Furthermore, it will be paramount in the future to assess whether EXO1 can be identified as a new FA gene in unassigned FA patients or if there are characterized FA patients bearing EXO1 mutations as a driver of the disease. In this case, these patients might be susceptible to endogenous formaldehyde levels, and their clinical outcomes should be closely monitored."

Reviewer #3 (Remarks to the Author):

The authors have performed a number of new experiments that have helped to elucidate the role of EXO1 in FA repair and clarify its role in relation to the Fanconi pathway. These have improved the manuscript and provided some new insights. I have only a few minor comments.

REPLY: Thank you. We would like to thank the reviewer for his/her comments which improved the manuscript.

1. Related to the new experiments that were performed following another reviewer's suggestion, I would like to point out that a protective effect of EXO1 loss on forks was reported previously (Rein et al, NAR 2015) after HU and CPT treatment. It was also reported that MUS81 co-depletion did not alter the sensitivity of EXO1 null cells to CPT. Given an opposite effect is observed with FA treatment, it would indicate there is some degree of specificity for the damage type that may be interesting to point out given the somewhat contradictory data addressing the role of EXO1 in damage sensitivity. The Rein paper also reports no IR sensitivity to EXO1 null cells, consistent with data shown in the paper and related to another reviewer comment regarding how "surprising" it is- it is published and could be referenced.

REPLY: We added the sentence in the discussion: "Of note, it was reported that Mus81 depletion did not alter the sensitivity of mouse EXO1 null cells to low-dose CPT treatment³⁶. Since a rescue of γ H2AX accumulation in EXO1 KO cells is observed with formaldehyde treatment, it indicates some degree of specificity for the damage type." Also : "Such a protective effect of EXO1 loss on DNA replication forks was reported previously after HU and CPT treatment³⁶." We added the reference to IR sensitivity in the main text: "The latter observation is consistent with the absence of significant IR sensitivity in mouse EXO1 KO cells² and human EXO1 KO clones compared to WT cells (Suppl. Figure 5a)."

2. In our hands, the RADAR assay has been reliable in identifying TOP1 and TOP2ccs, but we and others are using a slot blotter to concentrate the signal. There are likely less DPCs with FA than CPT or Etoposide treatment. It would be useful to make sure that the data throughout the paper is labeled as TOP1cc and not TOPcc for clarity. In the rebuttal, the authors state "EXO1 deletion leads to DPC accumulation using two methods (Topcc detection and RADAR assay)" but the graph in Supp Figure 8b does not appear to show any difference in the RADAR assay. Please clarify this result- it was seen with 2 methods or

it was seen with 1 method?

REPLY: We labelled the figures with TOP1cc and updated the text accordingly. Two methods were used to investigate DPC accumulation in EXO1 deficient cells. In our hands, Top1cc detection was the most reliable and showed significance. The RADAR assay shows the same tendency looking at the intensity of the smear on the SDS-PAGE. Again, In the future, we look forward to using a more sensitive method for DPC measurement.

1. Kratz, K. *et al.* FANCD2-Associated Nuclease 1 Partially Compensates for the Lack of Exonuclease 1 in Mismatch Repair. *Mol Cell Biol* **41**, e0030321 (2021).
2. Rein, K. *et al.* EXO1 is critical for embryogenesis and the DNA damage response in mice with a hypomorphic Nbs1 allele. *Nucleic Acids Res* **43**, 7371-7387 (2015).